# Residual Alignment:
# Uncovering the Mechanisms of Residual Networks

**Jianing Li**
University of Toronto
jrobert.li@mail.utoronto.ca

**Vardan Papyan**
University of Toronto
vardan.papyan@utoronto.ca

## Abstract

The ResNet architecture has been widely adopted in deep learning due to its significant boost to performance through the use of simple skip connections, yet the underlying mechanisms leading to its success remain largely unknown. In this paper, we conduct a thorough empirical study of the ResNet architecture in classification tasks by linearizing its constituent residual blocks using Residual Jacobians and measuring their singular value decompositions. Our measurements (code) reveal a process called Residual Alignment (RA) characterized by four properties:

(RA1)  intermediate representations of a given input are *equispaced* on a *line*, embedded in high dimensional space, as observed by Gai and Zhang [2021];

(RA2)  top left and right singular vectors of Residual Jacobians align with each other and across different depths;

(RA3)  Residual Jacobians are at most rank $C$ for fully-connected ResNets, where $C$ is the number of classes; and

(RA4)  top singular values of Residual Jacobians scale inversely with depth.

RA consistently occurs in models that generalize well, in both fully-connected and convolutional architectures, across various depths and widths, for varying numbers of classes, on all tested benchmark datasets, but ceases to occur once the skip connections are removed. It also provably occurs in a novel mathematical model we propose. This phenomenon reveals a strong alignment between residual branches of a ResNet (RA2+4), imparting a highly rigid geometric structure to the intermediate representations as they progress *linearly* through the network (RA1) up to the final layer, where they undergo Neural Collapse.

## 1 Introduction

### 1.1 Background

The Residual Network (ResNet) architecture [He et al., 2016a], a special case of Highway Networks [Srivastava et al., 2015], has taken the field of deep learning by storm, since its proposal in 2015. The incredibly simple architectural modification of adding a skip connection, spanning a set of layers, has become a go-to architectural choice for deep learning researchers and practitioners alike. Initially applied in computer vision, it has since been integrated as a crucial component in biomedical imaging and generative models via the U-Net [Ronneberger et al., 2015], natural language processing through the Transformer [Vaswani et al., 2017], and reinforcement learning as seen in the case of AlphaGo Zero [Silver et al., 2017].

The ResNet architecture first passes an input $x$ through initial layers $\mathcal{I}$ (comprising a sequence of convolution, batch normalization, and ReLU operations), depending on trainable parameters $\mathcal{W}_0$, to

37th Conference on Neural Information Processing Systems (NeurIPS 2023).

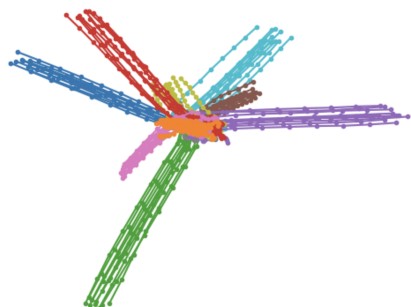

Figure 1: **Visualization of Residual Alignment.** Intermediate representations of a ResNet34[1], trained on CIFAR10, are projected onto two random vectors. Representations of each individual image are color-coded based on its true label and connected to form a trajectory, so as to showcase their progression throughout the network. Notice the *linear* arrangement of intermediate representations along with *equidistant* spacing between representations corresponding to consecutive layers (RA1) . Our work shows, this phenomenon results from the *alignment* of top singular vectors of Residual Jacobians (RA2) and the *inverse scaling* of top singular values with depth (RA4) . It is also noteworthy that the magnitudes of class means significantly increase with depth compared to the within-class variability, indicating the representations undergo layer-wise Neural Collapse [Papyan, 2020, Galanti et al., 2022, He and Su, 2022, Li et al., 2023].

generate an initial representation

$$h_1 = \mathcal{I}(x; \mathcal{W}_0) \in \mathbb{R}^D.$$

This is then processed through a sequence of residual blocks

$$h_{i+1} = \sigma(h_i + \mathcal{F}(h_i; \mathcal{W}_i)), \quad 1 \le i \le L,$$

each refining the previous layer's representation, $h_i \in \mathbb{R}^d$, through a simple residual branch

$$\mathcal{F}(h_i; \mathcal{W}_i) : \mathbb{R}^D \to \mathbb{R}^D,$$

which depends on trainable parameters $\mathcal{W}_i$ and an element-wise non-linearity, $\sigma$, which is simply an identity mapping, in the case of pre-activation ResNets [He et al., 2016b]. The final block's output, $h_{L+1}$, is fed to a classifier,

$$\mathcal{C}(h_{L+1}; \mathcal{W}_{L+1}) : \mathbb{R}^D \to \mathbb{R}^C,$$

depending on trainable parameters $\mathcal{W}_{L+1}$.

In the original ResNet architecture, $\mathcal{F}$ and $\mathcal{C}$ are compositions of linear transformations, element-wise nonlinearities, and normalization layers. The WideResNet architecture, by Zagoruyko and Komodakis [2016], further incorporates dropout layers into $\mathcal{F}$. Yet more changes are made in ResNext [Xie et al., 2017], where $\mathcal{F}$ is computed from the summation of parallel computational branches.

Training a ResNet for classification involves minimizing a loss on a training dataset, $\{x_n, y_n\}_{n=1}^N$, consisting of inputs $x_n$ and labels $y_n$ plus a weight decay term

$$\underset{\{\mathcal{W}_i\}_{i=1}^{L+1}}{\text{minimize}} \ \frac{1}{N} \sum_{n=1}^N \mathcal{L}(f(x_n; \mathcal{W}), y_n) + \frac{\lambda}{2} \|\mathcal{W}\|_2^2, \tag{1}$$

where $f(x_n; \mathcal{W})$ are the outputs of the classifier $\mathcal{C}$, also called logits, and $\mathcal{W}$ are all the parameters.

### 1.2 Problem Statement

The significant improvement in performance, achieved through the simple addition of a skip connection, has generated interest in understanding the underlying mechanisms of ResNets. Despite this, to this date, no definitive theory or explanation has been widely accepted by the deep learning community for the success of ResNets.

---

[1]ResNet34 has $(34 - 2)/2 = 16$ residual blocks since each residual block consists of two weight matrices and two layers are subtracted—one due to the initial convolution and the other at the final classification layer. Therefore, we would expect to see at most 16 dots. In the plot, we can count around 10 dots but the rest are too cluttered around the center. This is in agreement with Figure C.4 in the Appendix showing the inverse scaling of the top singular value occurs for roughly the last blocks, which correspond to the visible dots.

## 1.3 Method Overview

In this paper, we conduct a thorough empirical study of the ResNet architecture and its constituent residual blocks with the aim of investigating the characteristics of an individual residual block and the relationship between any pair of them. As the residual blocks are nonlinear functions of their inputs, we examine their linearizations through the *Residual Jacobian matrices*[2]:

$$\sigma'(h_i + \mathcal{F}(h_i; \mathcal{W}_i))\frac{\partial \mathcal{F}(h_i; \mathcal{W}_i)}{\partial h_i} \in \mathbb{R}^{D \times D}.$$

Following Equation (1.1), these correspond to the derivative of the residual block with respect to its input, *excluding* the contribution from the skip connection, $\sigma'(h_i + \mathcal{F}(h_i; \mathcal{W}_i)) \in \mathbb{R}^{D \times D}$. In the case of a pre-activation ResNet, these are simply equal to the derivative of the residual branch with respect to its input, i.e., $\partial \mathcal{F}(h_i; \mathcal{W}_i)/\partial h_i$. Since the Residual Jacobians are high-dimensional matrices, and likely contain meaningful information only in some of their subspaces, we measure their singular value decomposition (SVD), given by $J_i = U_i S_i V_i^\top$, where $U_i$ and $V_i$ are the respective left and right singular vectors, and $S_i$ is the singular value matrix.

## 1.4 Contributions

We discover a phenomenon called *Residual Alignment (RA)*, consistently occurring in ResNet models that generalize well, characterized by four properties[3]:

(RA1) Intermediate representations of a given input are *equispaced* on a *line*, embedded in high dimensional space, as shown in Figure 1 and observed by Gai and Zhang [2021];

(RA2) Top left and right singular vectors of Residual Jacobians align with each other and across different depths, as observed in Figure 2;

(RA3) Residual Jacobians are at most rank $C$ for fully-connected ResNets, where $C$ is the number of classes, as illustrated in Figures 3 and 4; and

(RA4) Top singular values of Residual Jacobians scale inversely proportional with depth, as depicted in Figure 4.

The properties are interrelated and, in fact, (RA1) can be logically derived from the other properties, as demonstrated in Section 3.

As a further contribution, in section B of the Appendix we prove theoretically the emergence of RA under the setting of binary classification with cross-entropy loss. Our proof relies on a novel mathematical abstraction called the *Unconstrained Jacobians Model*, in which the Residual Jacobians are optimized directly, so as to minimize the loss, and are not constrained by the architecture and parameters of the residual branches. This mathematical abstraction is motivated by recent theoretical works on Neural Collapse, as discussed in Section 5.

## 1.5 Results Summary

Our empirical investigation, presented in the main text and the Appendix, consistently identifies RA across an extensive range of:

**Architectures:** standard ResNets (convolutional layers with progressively increasing channels, interspersed with downsampling layers) as well as simpler designs (fully-connected layers);

**Datasets:** MNIST, FashionMNIST, CIFAR10, CIFAR100, and ImageNette [Howard]; and

**Hyperparameters:** network depth and width.

In addition to our main findings, we include an experiment showing the co-occurrence of (RA) and Neural Collapse in Figure 7. We also performed three counterfactual experiments that show:

---

[2]For a Type 1 model, $D = 512$ where 512 is the width of the network. For a Type 2 model, $D = 16 \times 16 \times 64 = 16384$ because the representations have 64 channels and a spatial dimension resolution of 16.

[3]These properties hold once the Jacobians are evaluated on the training data. If evaluated on other data, different phenomena might emerge. For example, we observed that the growth rate in (RA4) becomes exponential once the Jacobians are evaluated on the classification boundary, where the logits are close to zero.

1. If the number of classes in the dataset is increased, the singular vector alignment occurs in a higher dimensional subspace (Figure 3);

2. If stochastic depth [Huang et al., 2016] is incorporated, the singular vector alignment is amplified (Figure 5); and

3. If the skip connections are removed, RA does not occur (Figure 6);

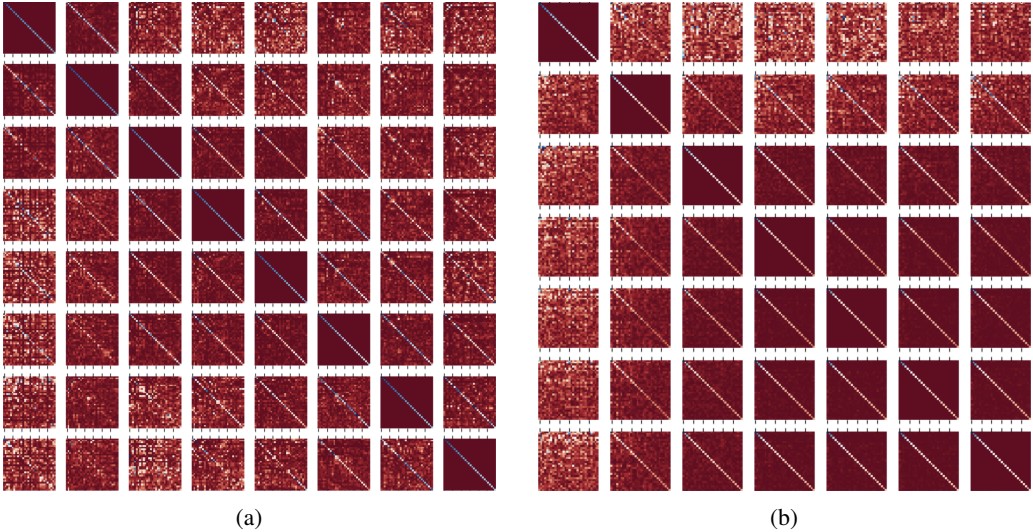

<table>
<tr><td>(a)</td><td>(b)</td></tr>
</table>

Figure 2: (RA2) : **Top singular vectors of Residual Jacobians align.** Subfigure 2a and Subfigure 2b present the alignment of the first 8 blocks and the last 7 blocks, respectively, for a ResNet34 trained on CIFAR100 (Type 3 model in Section 2.1) forwarding a single randomly sampled input. Each subplot $(i, j)$ illustrates the matrix $U_{j,30}^\top J_i V_{j,30}$, where $U_{j,30}$ and $V_{j,30}$ denote the top-30 left and right singular vectors of the Residual Jacobian $J_j$, respectively, and $i, j$ are the indices of the residual blocks, i.e., their depth. A distinct diagonal line of intense pixels is apparent in almost every subplot, signifying that the top singular vectors of $J_j$ diagonalize $J_i$. In simpler terms, this means that the top singular vectors of $J_i$ and $J_j$ align and (RA2) holds. This pattern persists when $V_{j,30}^\top J_i U_{j,30}$ is plotted, instead of $U_{j,30}^\top J_i V_{j,30}$, further confirming that the top left and right singular vectors align in accordance with (RA2) . Additional visualizations of both matrices, across various models and datasets, are available in subsections C.2 and C.3 of the Appendix. It is crucial to highlight that no alignment exists between the Jacobians at initialization, and the alignment emerges during training.

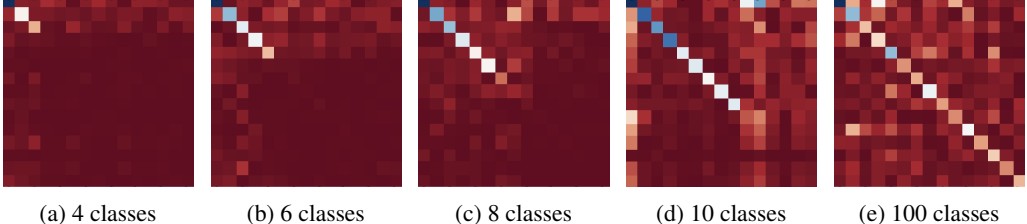

<table>
<tr><td>(a) 4 classes</td><td>(b) 6 classes</td><td>(c) 8 classes</td><td>(d) 10 classes</td><td>(e) 100 classes</td></tr>
</table>

Figure 3: (RA3) : **Singular vector alignment occurs in subspace of rank $\leq$ C.** The figure presents a sequence of subplots that illustrate the matrix $U_{16,10}^\top J_9 V_{16,10}$. Here, $J_9$ represents the 9-th Residual Jacobian, while $U_{16,10}$ and $V_{16,10}$ correspond to the leading 10 left and right singular vectors, respectively, of the 16-th Residual Jacobian, $J_{16}$. These calculations are based on ResNet34 models (Type 1 model in Section 2.1). These models have been trained on specific subsets of the CIFAR10 dataset, comprising of 4, 6, and 8 classes, as well as the complete CIFAR10 and CIFAR100 datasets. Each result is presented in the corresponding Subfigures 3a, 3b, 3c, 3d, and 3e. As the number of classes increases, the alignment of singular vectors occurs in an increasingly higher-dimensional subspace.

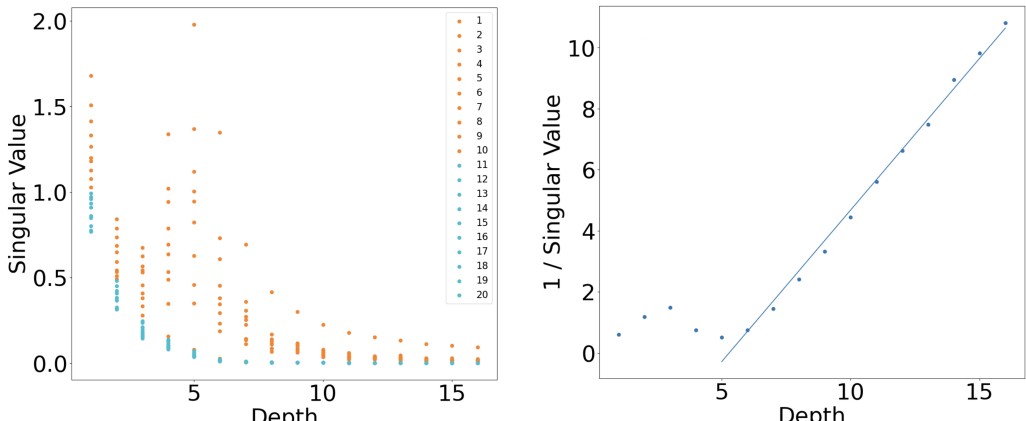

(a) (RA3) : **Residual Jacobians are at most rank** $C$. The top-10 singular values (orange) noticeably surpass the next 10 (blue), indicating that the Residual Jacobians rank is less than $C$.

(b) (RA4) : **Top singular value** $\approx 1/$**depth.** Post block 5, the reciprocal of the top singular value scales linearly with depth **and** with slope 0.992, indicating a singular value close to $1/i$ for the $i$-th block after RA begins.

Figure 4: Depiction of Residual Jacobian singular values for ResNet34 trained on CIFAR10 (Type 1 model in Section 2.1). Subfigure 4a shows the top 20 singular values of Residual Jacobians, while Subfigure 4b illustrates the inverse scaling of the top 1 values. More singular value plots, from diverse models and datasets, are available in subsection C.4 of the Appendix.

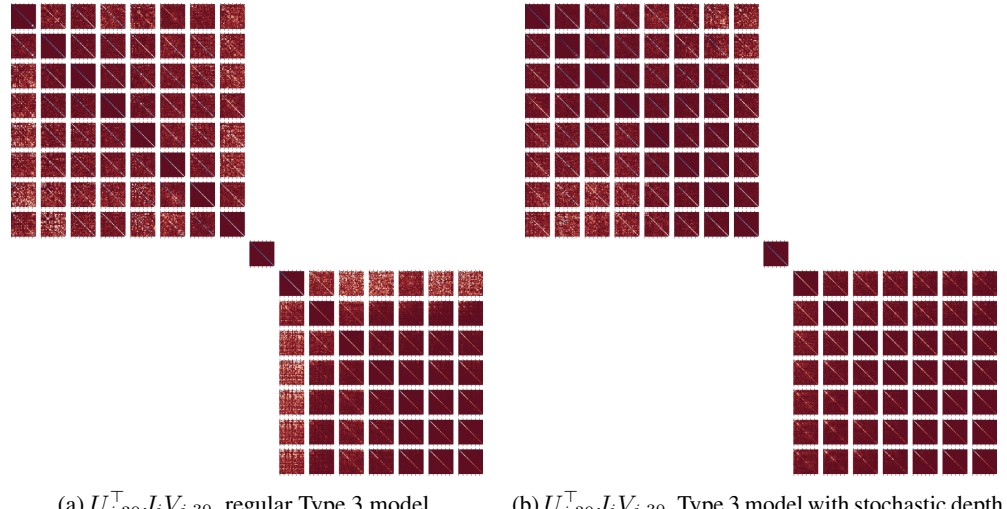

(a) $U_{j,30}^{\top} J_i V_{j,30}$, regular Type 3 model.

(b) $U_{j,30}^{\top} J_i V_{j,30}$, Type 3 model with stochastic depth.

Figure 5: **Stochastic depth amplifies singular vector alignment.** A comparison of (RA2) for two Type 3 models trained on CIFAR10 over 50 epochs: one employing the stochastic depth technique (with a drop probability of 0.3 for skipping residual blocks during training) and the other without it.

Table 1: (RA) **occurs in models that generalize well.** The table displays the test accuracies of models trained to study (RA) . Our reported accuracies closely align with those presented in Table 1 of Papyan et al. [2020], indicating that (RA) is observed in extensively trained models that exhibit strong performance in terms of test accuracy.

| Model | MNIST | FashionMNIST | CIFAR10 | CIFAR100 | ImageNette |
|-------|-------|--------------|---------|----------|------------|
| Type 1 | 98.9 | 90.9 | 58.3 | 30.4 | 42.6 |
| Type 2 | 99.5 | 92.0 | 88.5 | 54.6 | 67.9 |
| Type 3 | 99.6 | 92.8 | 87.3 | 62.6 | 64.2 |

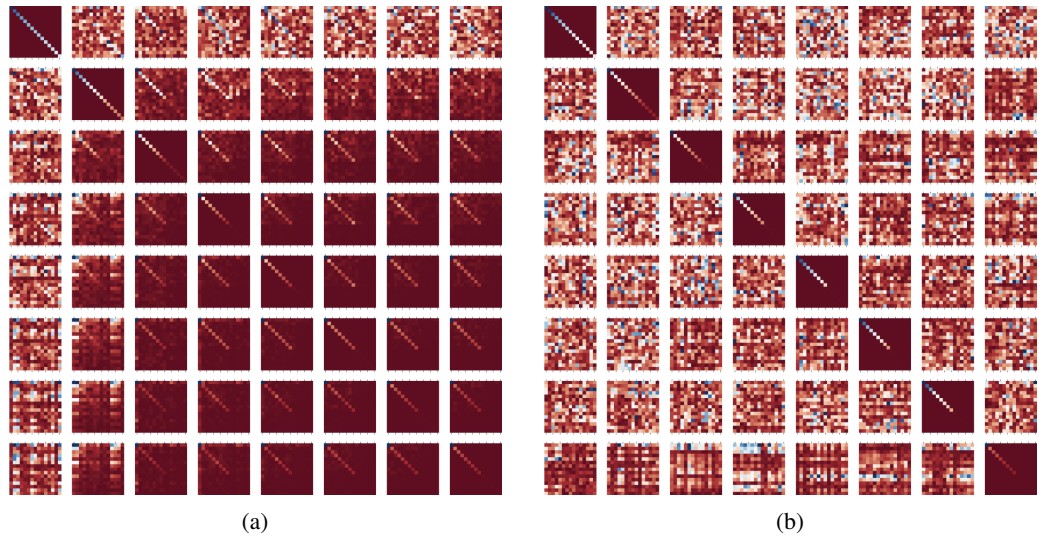

|       |       |
| :---: | :---: |
| (a)   | (b)   |

Figure 6: **Skip connections cause Residual Alignment.** The experiment depicted in Figure 2 is replicated using two additional models. The first is a ResNet18 trained on CIFAR10 (Type 4a model in Section 2.1), with results showcased in Subfigure 6a. The second is a ResNet18 *without skip connections* (Type 4b model in Section 2.1), with results displayed in Subfigure 6b. When the skip connections are removed, the top singular vectors no longer align. Alignment is visible in the diagonal subplots of Subfigure 6b, as each Residual Jacobian is diagonalized by its own singular vectors.

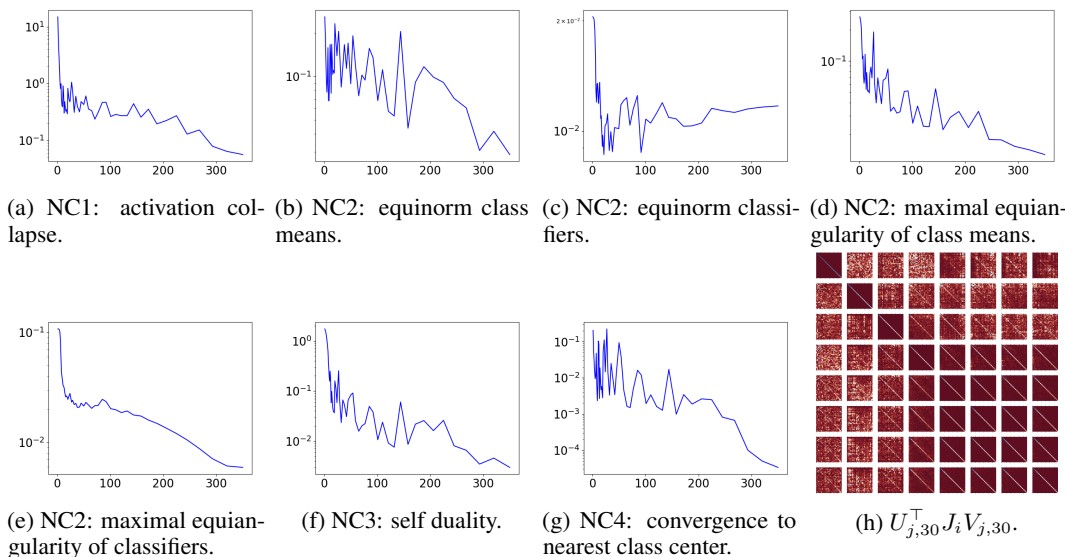

(a) NC1: activation collapse.

(b) NC2: equinorm class means.

(c) NC2: equinorm classifiers.

(d) NC2: maximal equiangularity of class means.

(e) NC2: maximal equiangularity of classifiers.

(f) NC3: self duality.

(g) NC4: convergence to nearest class center.

(h) $U_{j,30}^{\top} J_i V_{j,30}$.

Figure 7: **Co-occurrence of Residual Alignment and Neural Collapse.** The sub-figures display the progression of Neural Collapse metrics for a Type 4a model throughout 350 training epochs on the MNIST dataset as well as the emergence of Residual Alignment at the end of the training process.

## 2 Methods

### 2.1 Networks

We train 5 types of models:

**Type 1** models consist of 16 basic residual blocks, each containing two fully-connected layers of dimension $D{=}512$;

**Type 2** models consist of 16 basic residual blocks, each containing two convolutional layers with 64 channels, operating on a tensor with a $16 \times 16$ spatial resolution;

**Type 3** models are Type 2 models, but the last 8 basic residual blocks [4] have 128 channels operating on an $8 \times 8$ spatial resolution instead;

**Type 4** models are Type 1 models with $8$ instead of $16$ residual blocks. Type 4a models have skip connections and Type 4b models do not. These models serve the purpose of verifying that the residual connection causes RA, and RA does not occur once they are removed. [5];

**Type 5** models are obtained from Type 2 models in the same way that Type 4 models are obtained from Type 1 models. We again have Type 5a and Type 5b, denoting models with and without skip connections, respectively.

In our approach, we have deliberately chosen to use two spatial resolutions instead of the more common three. This decision allows us to have a greater number of residual blocks with the same spatial dimension.

## 2.2 Datasets

We train these models on the MNIST, FashionMNIST, CIFAR10, CIFAR100, and ImageNette [Howard] datasets. To explore the relationship between RA and the number of classes, we have trained additional models on subsets of these datasets, specifically with 2, 4, 6, and 8 classes.

## 2.3 Optimization

We train for $500$ epochs, using the SGD optimizer, with a batch size of $128$, an initial learning rate of $0.1$, the cosine learning rate scheduler [Loshchilov and Hutter, 2016], and a weight decay of $1e{-}1$ for convolutional models and $1e{-}2$ or $5e{-}2$ for fully-connected models. A large weight decay does not negatively impact the generalization performance, as seen in Table 1, but it does strengthen RA.

## 2.4 Randomized SVD

The Residual Jacobian matrix of a *convolutional* residual block with 64 channels operating on a spatial dimension of $16 \times 16$ is of size $16384 \times 16384$. Given that SVD scales cubically with the matrix side length, it becomes essential to leverage a more efficient algorithm to expedite computation and mitigate memory consumption. Furthermore, it is worth noting that we do not require the full SVD of the matrix. Our objective is to obtain the top singular values to confirm the trend in (RA3) and (RA4) and the top singular vectors to validate the alignment in (RA2).

We leverage Algorithm 971 [Li et al., 2017], a randomized SVD algorithm, to accurately compute the desired top singular values and vectors. This method involves a series of matrix multiplications and QR decompositions, requiring a runtime of $\mathcal{O}(TD^2K + TK^3)$ instead of $\mathcal{O}(D^3)$, where $T$ is the number of algorithm iterations, $D$ is the matrix side length, and $K$ is the number of top singular vectors. In our implementation, the $T$ is set to 20, and the $K$ ranges from 10 to 30, depending on the specific experiment. Through extensive examination, we have verified that this algorithm reliably provides singular values and vectors of sufficient accuracy.

## 3 Results

### 3.1 Empirical Results

The empirical results supporting (RA) are found in Table 1 and Figures 1, 2, and 4. The counterfactual experiments are in Figures 3, 5, and 6. We also include an experiment showing the co-occurrence of (RA) and Neural Collapse in Figure 7. Experimental descriptions and discussions are within the captions. Additional supporting data can be found in the Appendix.

---

[4] The 9th residual block downsamples the representation through a projection shortcut [He et al., 2016a].

[5] We use 8 blocks because the optimization fails to converge when using 16 blocks and no skip connections

## 3.2 (RA2+3+4) Imply (RA1)

As mentioned in the introduction, the properties of (RA) are interconnected, and this relationship is demonstrated through the following theorem:

**Theorem 3.1.** *For binary classification, in a pre-activation ResNet, assuming the Jacobian linearizations are exact and satisfy (RA2+3+4), then (RA1) holds for the intermediate representations.*

The proof of this Theorem is deferred to section A of the Appendix.

## 3.3 Unconstrained Jacobians Model Leads to RA

We propose the following abstraction of the optimization problem in Equation (1).

**Definition 3.2 (Unconstrained Jacobians Model).** Given a fixed input $\Delta_x \in \mathbb{R}^D$ and its label $y \in \{+1, -1\}$, find matrices $J_i \in \mathbb{R}^{D \times D}$, $1 \leq i \leq L$, and vector $w \in \mathbb{R}^D$ that

$$\underset{w, \{J_i\}_{i=1}^{L}}{\text{minimize}} \quad \mathcal{L}(w^\top \prod_{i=1}^{L}(I + J_i)\Delta_x, y) + \frac{\lambda}{2}\sum_{i=1}^{L}\|J_i\|_F^2 + \frac{\lambda}{2}\|w\|_2^2.$$

In the problem described above, $J_i$ again represents the Residual Jacobian of the $i$-th residual branch and $w$ represents the classifier Jacobian,

$$\frac{\partial \mathcal{C}(h_{L+1}; \mathcal{W}_{L+1})}{\partial h_{L+1}}.$$

It is referred to as the "Unconstrained Jacobians Model" because the Jacobians are not restricted to any specific form and are not required to be realizable by a set of layers. In the Unconstrained Jacobians Model, the Jacobians are regularized through simple functions. However, in reality, weight decay, dropout, normalization layers, and parallel branches regularize the Jacobians in intricate ways that are hard to capture mathematically.

We prove in section B the Appendix the following theorem:

**Theorem 3.3.** *For binary classification, there exists a global optimum of the Unconstrained Jacobians Model where the top Jacobian singular vectors are aligned (RA2), all Jacobians are rank one, analogous to (RA3), and the top Jacobian singular values are equal, analogous to (RA4).*

Here, the top singular values are equal and not decaying as predicted by (RA4), because the Jacobians are evaluated on the classification boundary instead of on training examples. Therefore, the intermediate representations are not equispaced on a line, as predicted by (RA1) but are rather exponentially-spaced on a line.

## 4 Discussion

In this section, we pose research questions that emerge from the discovery of RA.

**Generalization** The discovery of RA offers a vivid analogy for comprehending generalization in deep learning. According to RA, we can envision a ResNet as a system of conveyor belts, where each conveyor carries representations of a specific class in a unified direction at a constant speed. Misclassification occurs when the representation mistakenly steps onto the wrong conveyor belt in the initial layers of the network. This perspective leads us to hypothesize that the first few layers of the network play a vital role in generalization, surpassing the significance of subsequent layers. Consequently, more research should be dedicated towards studying the dynamics of the initial layers.

**Neural Collapse** As mentioned in Figure 1, there is an intriguing pattern where models exhibit RA concurrently with layer-wise Neural Collapse [Papyan, 2020, Galanti et al., 2022, Li et al., 2023]. The concurrent manifestation of these two distinct events could possibly be more than mere coincidence. It would be interesting to explore if RA could shed light on phenomena related to layer-wise Neural Collapse such as the "Law of Data Separation" [He and Su, 2022].

**RA in Transformers**    We have empirically substantiated the occurrence of RA in ResNets. However, our study has not yet extended to Transformers, which also incorporate residual connections. An intriguing line of inquiry for future work would be to probe whether RA, or a phenomenon akin to it, manifests within these architectures.

**Recurrent Architectures Exhibiting RA**    Recurrent neural networks, Neural ODEs [Chen et al., 2018], and deep equilibrium models [Bai et al., 2019] iteratively apply a layer within a deep neural network. This leads to the following questions:

*Do the intermediate representations of these models exhibit RA? If not, can we propose a novel architecture that recurrently applies a computation but does exhibit RA?*

**Model Compression**    In our work, we demonstrate that the network can converge to a model that iteratively applies a computation, even without imposing explicit constraints, like the architectures in the previous subsection. This leads us to yet another question:

*Can we replicate the original network's performance by distilling the aligning layers into a single layer and iteratively applying it?*

**Regularization Techniques**    Existing regularization techniques, including layer permutation [Liaw et al., 2021] and structured dropout [Fan et al., 2019], should strengthen RA. A question arises:

*Could RA explain the success of these methods and do they indeed amplify RA?*

## 5    Related Work

**Linearization of Residual Jacobians**    Prior to our work, Rothauge et al. [2019] proposed the linearization of residual Jacobians and investigated the distribution of their singular values. Our work, however, concerns the discovery of the alignment between the Residual Jacobians, as well as its theoretical understanding.

**Generalization**    A thorough empirical investigation by Novak et al. [2018] found that the Frobenius norm of the input-output Jacobian of a network correlates well with generalization. Our research complements theirs by conducting both empirical and theoretical investigations on the *Residual Jacobians*, which form the input-output Jacobian of a ResNet.

**ResNets at Initialization**    Previous works have contributed significantly to the understanding of the properties of intermediate representations of randomly initialized ResNets, with studies such as Hayou [2022] exploring the infinite-width and finite-width regime and Li et al. [2021, 2022] investigating the infinite-depth and infinite-width regime. Our current research diverges from these previous works by specifically focusing on studying the Residual Jacobians of fully trained ResNets.

**Optimization Landscape**    Li et al. [2016] analyzed the Hessian of the loss function for a ResNet initialized with zero parameters. Additionally, Lu et al. [2020] use a mean-field analysis of ResNets to demonstrate convergence to a global minimum. Their analysis builds upon the observation that a ResNet is similar to a shallow network ensemble, first noted by Veit et al. [2016].

Rather than focusing on the optimization convergence properties of SGD, our goal is to examine the properties of the intermediate representations and Residual Jacobians that emerge during training.

**Neural Collapse**    Recent theoretical work by Mixon et al. [2020], Lu and Steinerberger [2020], E and Wojtowytsch [2020], Poggio and Liao [2020], Zhu et al. [2021], Han et al. [2021], Tirer and Bruna [2022], Wang et al., Kothapalli et al. [2022] analyzed the Neural Collapse phenomenon, discovered by Papyan, Han, and Donoho [2020], through the unconstrained features model and the layer-peeled model. In these, the assumption is made that the last-layer representations, which are fed to a classifier, have the freedom to move independently and are not constrained to be the output of a deep network. Our Unconstrained Jacobians Model takes inspiration from these mathematical models by abstracting away the complex Residual Jacobians and assuming they can be optimized directly.

**Unrolled Iterative Algorithm Perspective** Greff et al. [2016], Papyan et al. [2017], and Ebski et al. [2018] recognized ResNets as unrolled iterative algorithms performing iterative inference. We build upon this understanding by delving deeper into the empirical and theoretical relationships between the Residual Jacobians.

**Neural ODE** Chen et al. [2018] introduced the Neural ODE: a continuous-depth, tied-weights ResNet. Initially, it may seem unlikely that the iterations or Jacobians of a traditional ResNet would possess any characteristics associated with Neural ODEs. However, Sander et al. [2022] proved that assuming the initial loss is small and the initial parameters are smooth with depth, *linear* ResNets converge to a Neural ODE as the number of layers increases. Still, it is unclear if these findings extend to *nonlinear* ResNets trained on *real* data.

Our work, however, demonstrates that even in such cases the Residual Jacobians align in their top subspaces and, as a result, simple ODEs emerge within these subspaces. We also provide theoretical justification for this claim through the Unconstrained Jacobians Model.

**Optimal Transport** Gai and Zhang [2021] view ResNets as aiming to transport an input distribution to a label distribution, through a geodesic curve in the Wasserstein space, induced by the optimal transport map. They provide empirical evidence to support their claim, showing that intermediate representations are equidistant on a straight line induced by the optimal transport map, and comment: *"Though ResNet approximates the geodesic curve better than plain network, it may not be a perfect implement in high-dimensional space due to its layer-wise heterogeneity."*

Our research demonstrates that intermediate representations lie on a line as a result of the Jacobian singular vectors aligning (RA2) and the Jacobian singular values scaling inversely with depth (RA4) and, in fact, due to the *absence* of purported "layer-wise heterogeneity."

**Analysis of Deep Linear Networks** Mulayoff and Michaeli [2020] proved that training a deep *linear network*, with a *quadratic loss*, and *no weight decay*, necessarily converges to the flattest of all minima. At this optimum point, the spectral norm of the input-feature Jacobian increases exponentially with depth, and the singular vectors of consecutive weight matrices align.

Our theoretical study considers *ResNets*, trained with a *binary cross-entropy loss*, and *weight decay*. Instead of assuming that training has reached a flat minimum, we analyze the Unconstrained Jacobians Model, on the classification boundary in the input space, and prove phenomena that we have observed through experiments on *nonlinear* ResNets.

**Analysis of Weights** Cohen et al. [2021] studied the scaling behavior of trained *weights* in deep residual networks. Our study complements theirs by focusing on examining the *Residual Jacobians* of ResNet architectures.

## 6 Conclusion

In this paper, we offer a detailed empirical examination of the ResNet architecture, a model that has seen extensive application across a broad spectrum of deep learning domains. Our primary aim was to demystify the mechanisms that contribute to ResNet's remarkable success, a topic that, to date, remains an enigma within the deep learning community.

Our investigation has led to the discovery of a consistent phenomenon, which we have termed RA. The characteristics of RA were observed across a wide array of benchmark datasets, canonical architectures, and hyperparameters, demonstrating its general applicability. Moreover, we conducted counterfactual studies that underscored the critical role of skip connections in the emergence of RA and the effect of the number of classes. In an attempt to theoretically ground the emergence of RA, we proposed the use of an innovative mathematical abstraction – the Unconstrained Jacobians Model, specifically in the context of binary classification with cross-entropy loss.

Our exploration has not only shed light on the intricate mechanisms driving ResNets' performance but also points to connections with the recent phenomenon of layer-wise Neural Collapse. Furthermore, our findings pave the way for future research in the understanding of existing regularization methods, the design of novel architectures, the development of model compression techniques, and the theoretical investigation of generalization.

## Acknowledgments and Disclosure of Funding

We thank Kirill Serkh for inspiring discussions and feedback on the manuscript. We acknowledge the support of the Natural Sciences and Engineering Research Council of Canada (NSERC), [funding reference number 512236 and 512265]. This research was enabled in part by support provided by Compute Ontario (http://www.computeontario.ca) and Compute Canada (http://www.computecanada.ca).

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

# A (RA2+3+4) Imply (RA1)

**Theorem 3.1.** *For binary classification, in a pre-activation ResNet, assuming the Jacobian linearizations are exact and satisfy (RA2+3+4), then (RA1) holds for the intermediate representations.*

*Proof.* In a pre-activation ResNet,

$$h_{i+1} = h_i + \mathcal{F}(h_i; \mathcal{W}_i).$$

Since Jacobian linearizations are exact, we have:

$$h_{i+1} = (I + J_i)h_i.$$

Recall, the singular value decomposition of $J_i$ is given by

$$J_i = U_i S_i V_i^\top,$$

where $U_i$ and $V_i$ are the respective left and right singular vector matrices, and $S_i$ is the singular value matrix. Invoking (RA2) and (RA3), for some matrix $U_0$ and for any block $i$,

$$J_i = U_0 S_i U_0^\top,$$

and therefore

$$h_{i+1} = (I + U_0 S_i U_0^\top)h_i.$$

Applying recursively the above equality leads to

$$h_k = \left( \prod_{i=1}^{k-1} (I + U_0 S_i U_0^\top) \right) h_1 = U_0 \left( \prod_{i=1}^{k-1} (I + S_i) \right) U_0^\top h_1. \tag{2}$$

For binary classification, (RA3) implies the Jacobians are rank 1 and therefore

$$h_k = U_{0,1} \left( \prod_{i=1}^{k-1} (1 + S_{i,1}) \right) U_{0,1}^\top h_1 + (I - U_{0,1} U_{0,1}^\top)h_1,$$

where $U_{0,j}$ is the $j$th column of $U_0$. According to (RA4),

$$S_{i,1} = \frac{1}{i}$$

and

$$\prod_{i=1}^{k-1} (1 + S_{i,1}) = \prod_{i=1}^{k-1} (1 + 1/i) = k.$$

Substituting the above into Equation (2), we obtain

$$h_k = k U_{0,1} U_{0,1}^\top h_1 + (I - U_{0,1} U_{0,1}^\top)h_1 = (k-1) U_{0,1} U_{0,1}^\top h_1 + h_1.$$

This proves that the intermediate representations of a given input are *equispaced* on a *line* embedded in high dimensional space, i.e., (RA1).

$\square$

# B Unconstrained Jacobians Model Leads to RA

We start by providing motivation for the unconstrained Jacobians problem introduced in the main text. Assume a training example, $x$, is situated next to a point on the classification boundary, denoted by $x_{\mathrm{mid}}$, satisfying $f(x_{\mathrm{mid}}; \mathcal{W}) = 0$. Performing a Taylor expansion of the logits of $x$ around $x_{\mathrm{mid}}$ yields

$$f(x; \mathcal{W}) = f(x_{\mathrm{mid}}; \mathcal{W}) + \left. \frac{\partial f(x; \mathcal{W})}{\partial x} \right|_{x_{\mathrm{mid}}} \Delta_x + h(x, x_{\mathrm{mid}})$$

$$= \left. \frac{\partial f(x; \mathcal{W})}{\partial x} \right|_{x_{\mathrm{mid}}} \Delta_x + h(x, x_{\mathrm{mid}}), \tag{3}$$

where $\Delta_x = x - x_{\text{mid}}$ and $h(x, x_{\text{mid}})$ accounts for the approximation error, which is $\mathcal{O}(\|\Delta_x\|_2^2)$ and assumed to be negligible in our analysis.

Recall the loss associated with training a ResNet:

$$\underset{\{\mathcal{W}_i\}_{i=1}^{L+1}}{\text{minimize}} \ \frac{1}{N} \sum_{n=1}^{N} \mathcal{L}(f(x_n; \mathcal{W}), y_n) + \frac{\lambda}{2} \|\mathcal{W}\|_2^2. \tag{4}$$

Substituting Equation (3) into the above, neglecting the approximation error, and considering only the objective associated with the training sample $x$ and its label $y$, we get

$$\mathcal{L}\left(\left.\frac{\partial f(x; \mathcal{W})}{\partial x}\right|_{x_{\text{mid}}} \Delta_x, y\right) + \frac{\lambda}{2} \|\mathcal{W}\|_2^2.$$

By using the chain rule, we can then obtain the following:[6]

$$\mathcal{L}\left(\frac{\partial \mathcal{C}(h_{L+1}; \mathcal{W}_{L+1})}{\partial h_{L+1}} \prod_{i=1}^{L}\left(I + \frac{\partial \mathcal{F}(h_i; \mathcal{W}_i)}{\partial h_i}\right) \Delta_x, y\right) + \frac{\lambda}{2} \sum_{i=1}^{L} \|\mathcal{W}_i\|_F^2 + \frac{\lambda}{2} \|\mathcal{W}_{L+1}\|_F^2, \tag{5}$$

where all the Jacobians are evaluated at the point $x_{\text{mid}}$. This naturally gives rise to the following definition, as introduced in the main text.

**Definition 3.2 (Unconstrained Jacobians Model).** *Given a fixed input $\Delta_x \in \mathbb{R}^D$ and its label $y \in \{+1, -1\}$, find matrices $J_i \in \mathbb{R}^{D \times D}$, $1 \leq i \leq L$, and vector $w \in \mathbb{R}^D$ that*

$$\underset{w, \{J_i\}_{i=1}^{L}}{\text{minimize}} \quad \mathcal{L}(w^\top \prod_{i=1}^{L}(I + J_i)\Delta_x, y) + \frac{\lambda}{2} \sum_{i=1}^{L} \|J_i\|_F^2 + \frac{\lambda}{2} \|w\|_2^2.$$

In the main text, we stated the following theorem.

**Theorem 3.3.** *For binary classification, There exists a global optimum of the Unconstrained Jacobians Model where the top Jacobian singular vectors are aligned (RA2), all Jacobians are rank one, analogous to (RA3), and the top Jacobian singular values are equal, analogous to (RA4).*

*Proof.* Throughout the proof, we assume, without loss of generality, that the label is $y = 1$. Using the cyclic property of the trace, the logit, $w^\top \prod_{i=1}^{L}(I + J_i)\Delta_x$, equals

$$\text{tr}\left\{\Delta_x w^\top \prod_{i=1}^{L}(I + J_i)\right\}.$$

Denoting the power set of all natural numbers between 1 and $L$ by $\mathcal{P}(L)$, the above can be expressed as follows:

$$\sum_{s \in \mathcal{P}(L)} \text{tr}\left\{\Delta_x w^\top \prod_{i \in s} J_i\right\}.$$

Each element in the above summation can be upper bounded through the following theorem (a generalization of Von Neumann's trace inequality [Mirsky, 1975] to the product of more than two real matrices).

**Theorem 2** ([Miranda and Thompson, 1993]). *Let $A_1, \ldots, A_m$ be matrices with real entries.[7] Take the singular values of $A_j$ to be $s_1(A_j) \geq \cdots \geq s_n(A_j) > 0$, for $j = 1, \ldots, m$, and denote $S_j = \text{diag}(s_1(A_j), \ldots, s_n(A_j))$. Then, as the matrices $P_1, \ldots, P_m$ range over all possible rotations, i.e., the special orthogonal group $\text{SO}(n)$,*

$$\sup_{P_1, \ldots, P_m \in \text{SO}(n)} \text{tr}(A_1 P_1 \ldots A_m P_m)$$

$$= \sum_{i=1}^{n-1} \prod_{j=1}^{m} s_i(A_j) + [\text{sign} \det(A_1 \ldots A_m)] \prod_{j=1}^{m} s_n(A_j).$$

---

[6]For simplicity, we ignore the input transformation that maps the input $\Delta_x$ to the first representation $h_1 \in \mathbb{R}^D$ by simply assuming $\Delta_x \in \mathbb{R}^D$.

[7]Assume the convention that the singular values are positive.

*Moreover, assuming* $\operatorname{sign} \det(A_1 \dots A_m) = 1$,

$$\sup_{P_1,\dots,P_m\in \mathrm{SO}(n)} \operatorname{tr}(A_1 P_1 \dots A_m P_m) = \operatorname{tr}\left\{\prod_{i=1}^{m} S_i\right\}.$$

We will continue our proof using contradiction. Suppose all existing global optima of the unconstrained Jacobians problem consist of Jacobians that do not have aligning singular vectors, or do not have equal singular values, or are not rank 1. Then take any solution $\{J_i\}_{i=1}^{L}$ and $w$. Using the singular value decomposition, we have

$$J_i = U_i S_i V_i^\top, \quad \text{for } i = 1, \dots, L,$$

and

$$\Delta_x w^\top = U_{L+1} S_{L+1} V_{L+1}^\top.$$

Then Theorem 2 implies

$$\sum_{s\in\mathcal{P}(L)} \operatorname{tr}\left\{\Delta_x w^\top \prod_{i\in s} J_i\right\} \leq \sum_{s\in\mathcal{P}(L)} \operatorname{tr}\left\{S_{L+1} \prod_{i\in s} S_i\right\}$$

$$= \operatorname{tr}\left\{S_{L+1} \prod_{i=1}^{L}(I + S_i)\right\}.$$

For all $s \in \mathcal{P}(L)$, the inequality becomes equality once the singular vectors of all the Jacobians align with those of $\Delta_x w^\top$ and once the vector $w$ is chosen to be proportional to $\Delta_x$ (so that the matrix $\Delta_x w^\top$ is symmetric). The implication of the steps thus far is that one can increase, or at least keep constant the logit, and consequently reduce, or at least keep constant the loss by simply aligning the singular vectors of the Jacobians. In addition, since the regularization term $\|J_i\|_F^2 = \operatorname{tr}\{S_i^2\}$, this change of Jacobians does not affect the regularization terms.

Notice that $\Delta_x w^\top$ is a rank one matrix and so $S_{L+1}$ has a single non-zero diagonal entry. Furthermore, the matrices $S_i$, for $1 \leq i \leq L$, are all diagonal. As such, we can zero out all their other diagonal entries and leave a single non-zero entry at the location that $S_{L+1}$ has one, which does not affect the logits but reduces the regularization terms.

Using the inequality of arithmetic and geometric means on this only non-zero entry $s_i$ of every diagonal matrix $S_i$ gives

$$s_{L+1} \prod_{i=1}^{L}(1 + s_i) \leq s_{L+1}\left(1 + \frac{1}{L}\sum_{i=1}^{L} s_i\right)^L.$$

The implication of the above inequality is that, once the singular vectors of the Jacobians are aligned, one can further increase the logits and reduce the loss by averaging all the top singular values, $s_i$ for $1 \leq i \leq L$, and forcing them to be equal. Furthermore, since $\|J_i\|_F^2 = \operatorname{tr}\{S_i^2\} = s_i^2$ is convex, by Jensen's inequality, averaging the singular values only decreases the value of the Jacobian regularization.

All in all, we obtain higher, or at least no lower logit, and lower, or at least no higher loss when all singular vectors are aligned, all top singular values are equal and all other singular values are zero, which contradicts the statement that no global optima of the unconstrained Jacobians problem satisfies all of these conditions. $\square$

# C    Additional Empirical Evidence

## C.1    (RA1)

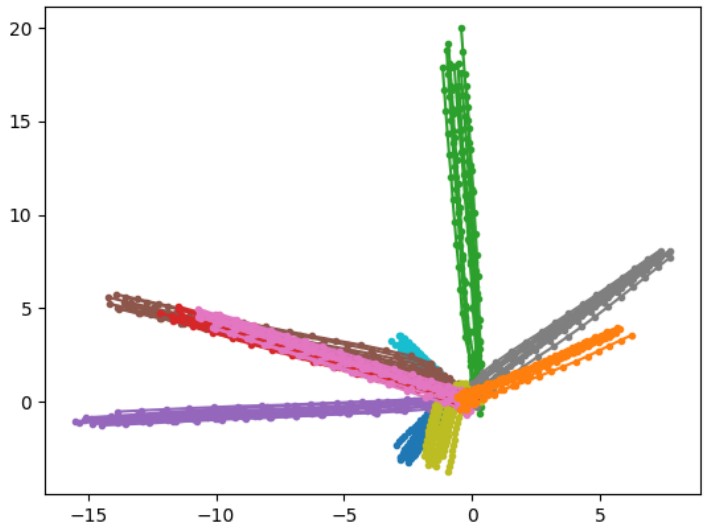

Figure 8: Fully-connected ResNet34 (Type 1 model) trained on MNIST.

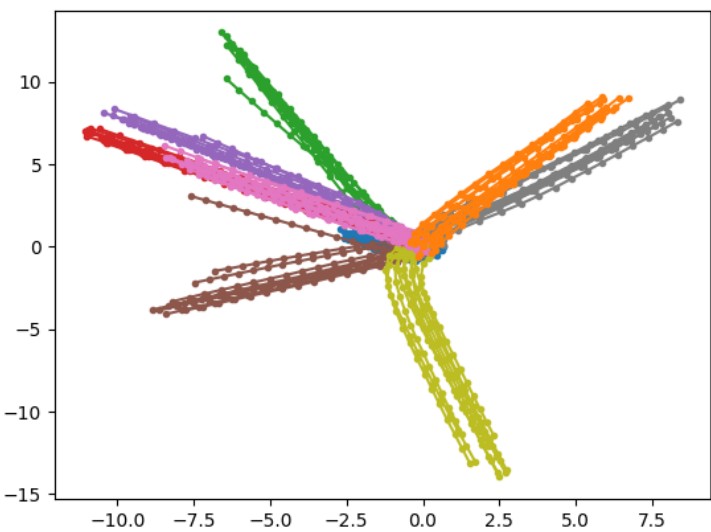

Figure 9: Fully-connected ResNet34 (Type 1 model) trained on FashionMNIST.

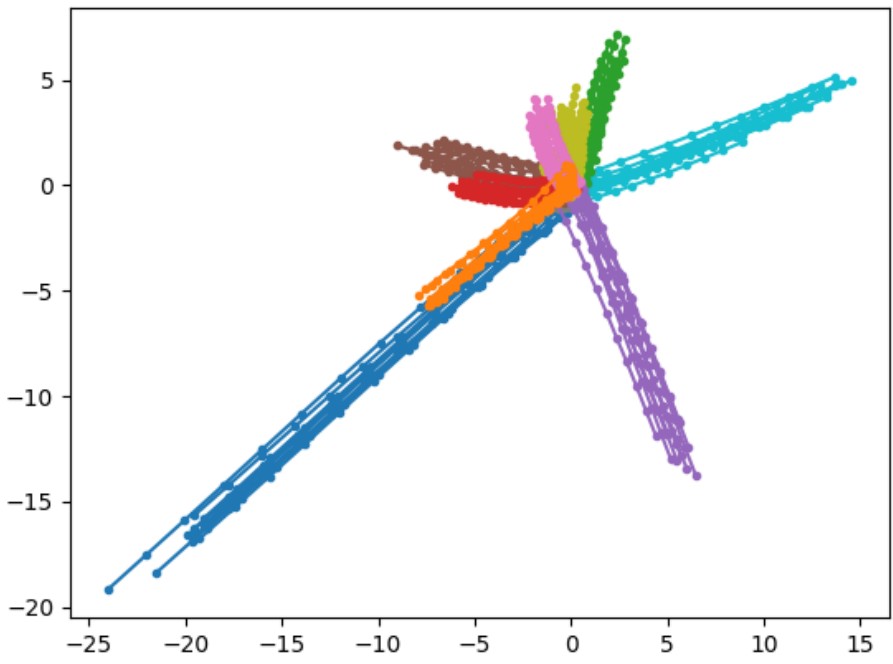

Figure 10: Fully-connected ResNet34 (Type 1 model) trained on CIFAR10.

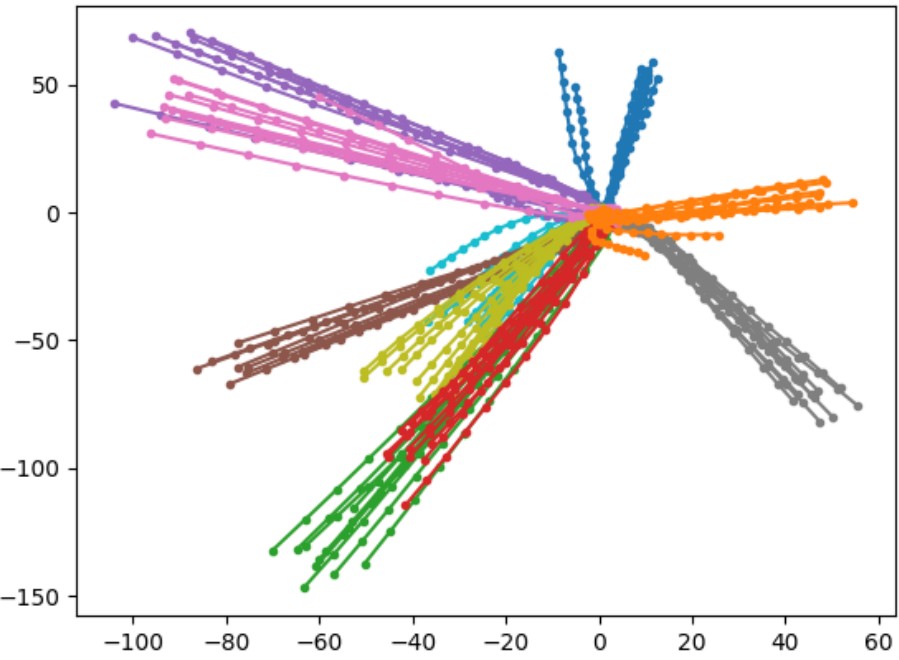

Figure 11: Convolutional ResNet34 (Type 2 model) trained on MNIST.

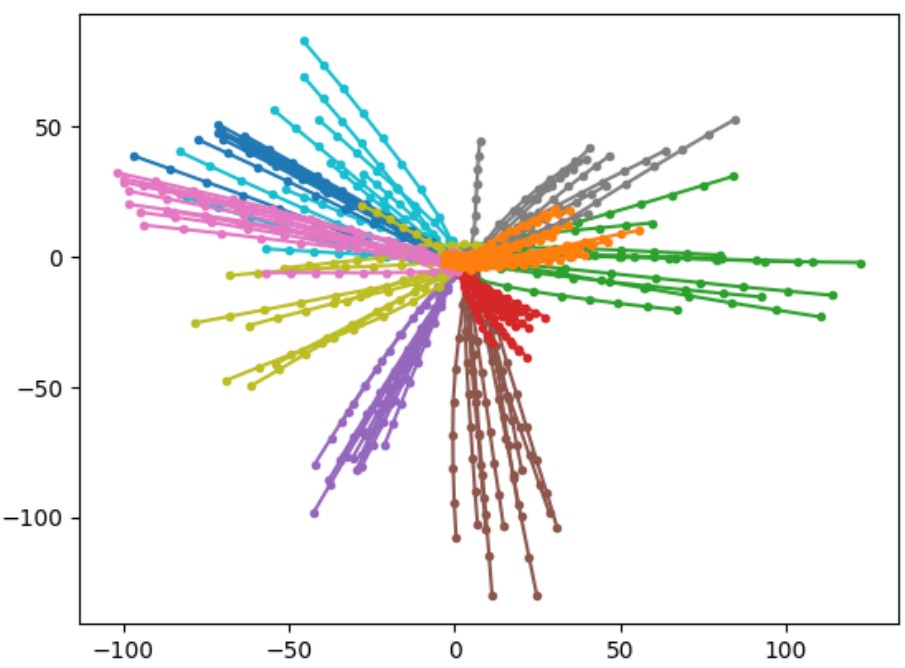

Figure 12: Convolutional ResNet34 (Type 2 model) trained on FashionMNIST.

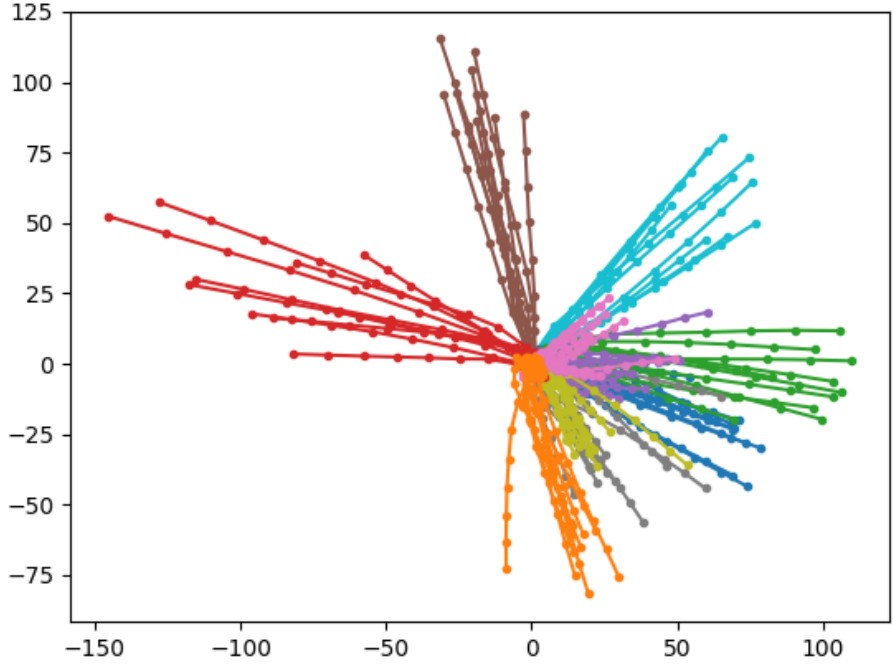

Figure 13: Convolutional ResNet34 (Type 2 model) trained on CIFAR10.

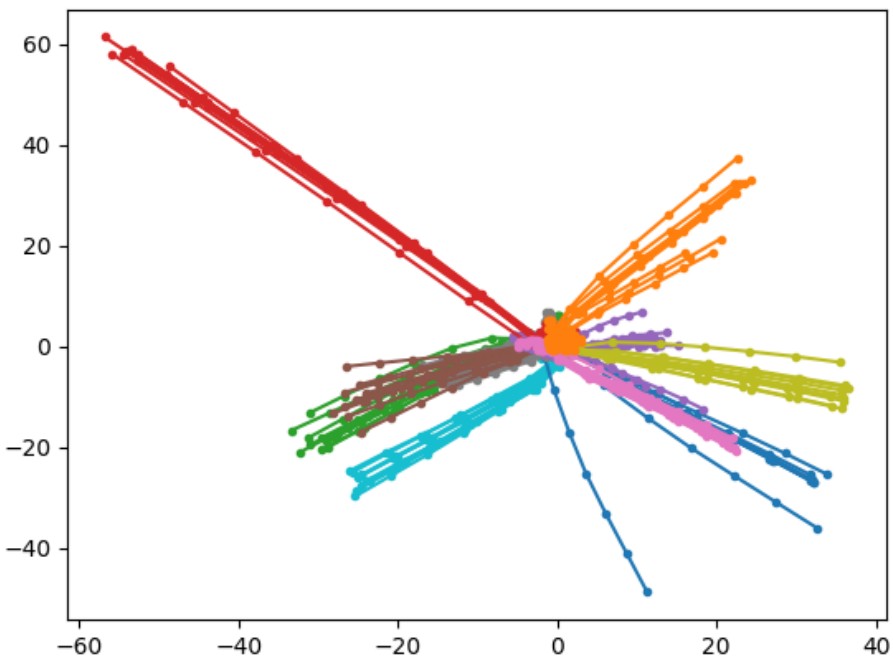

Figure 14: Convolutional ResNet34 with downsampling (Type 3 model) trained on MNIST.

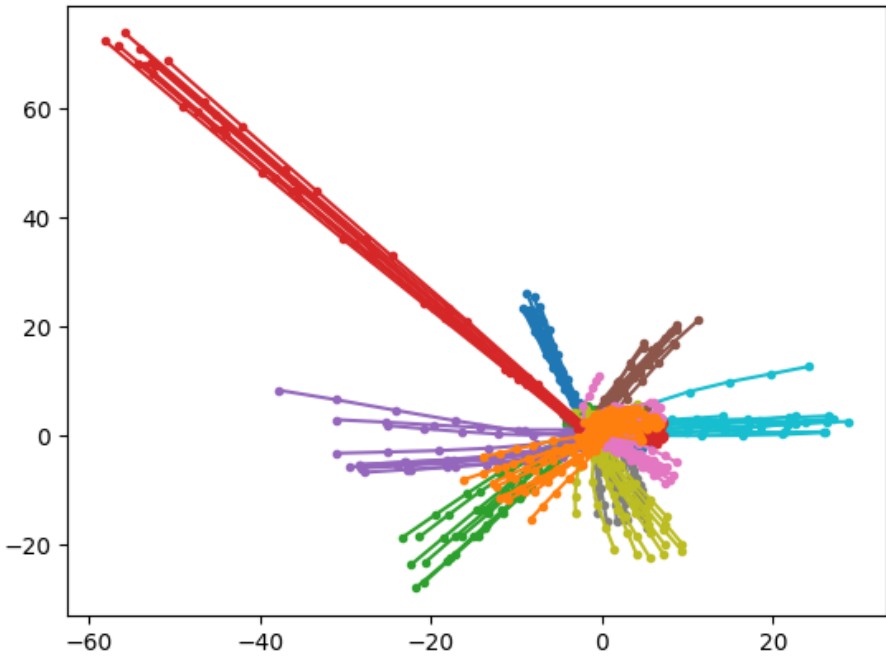

Figure 15: Convolutional ResNet34 with downsampling (Type 3 model) trained on FashionMNIST.

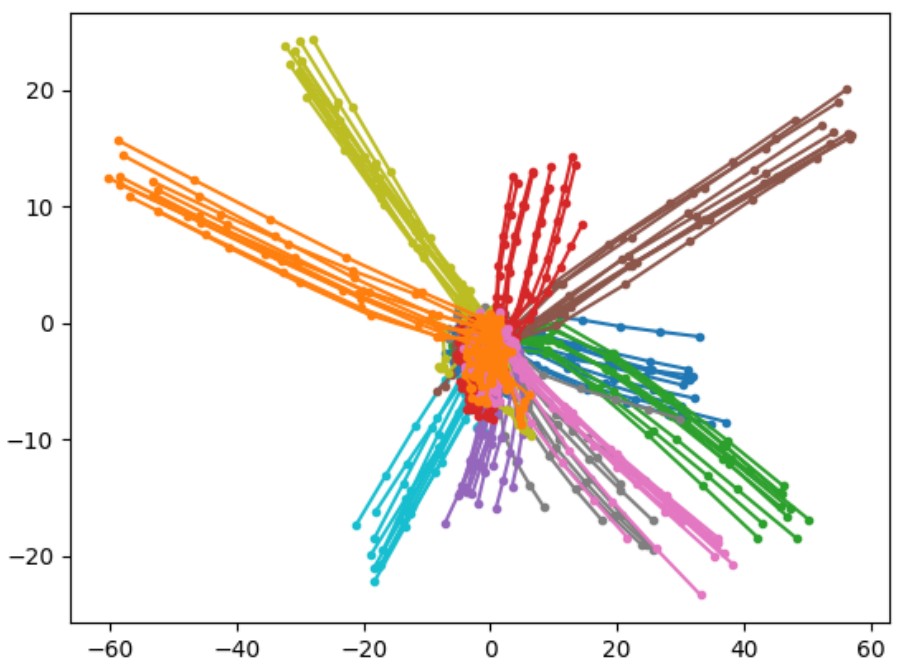

Figure 16: Convolutional ResNet34 with downsampling (Type 3 model) trained on CIFAR10.

## C.2 (RA2) : $U_{j,K}^\top J_i V_{j,K}$

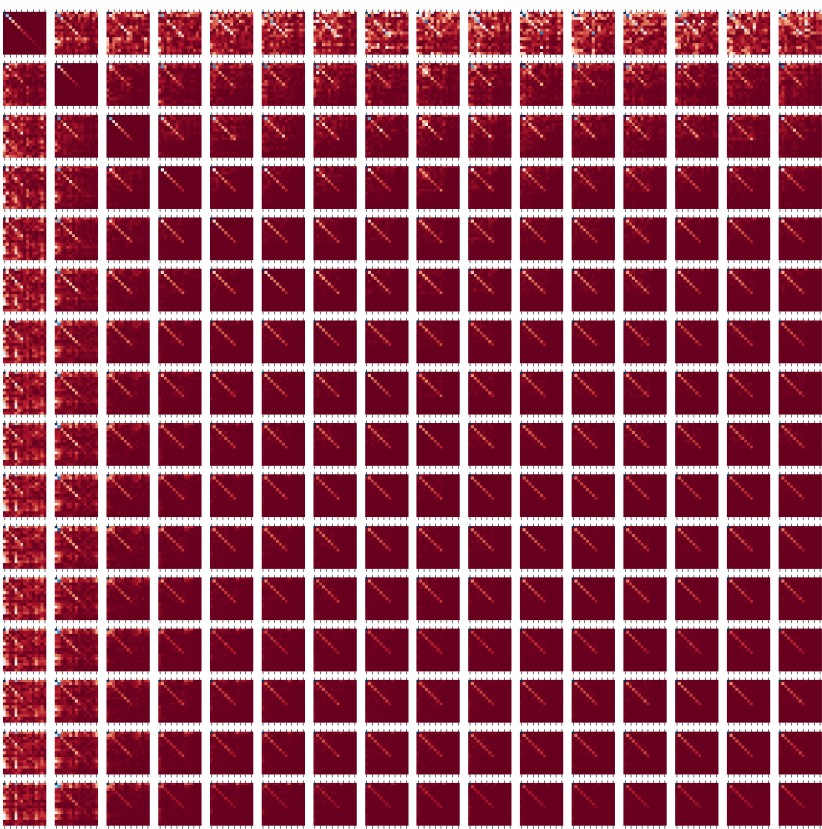

Figure 17: Fully-connected ResNet34 (Type 1 model) trained on MNIST.

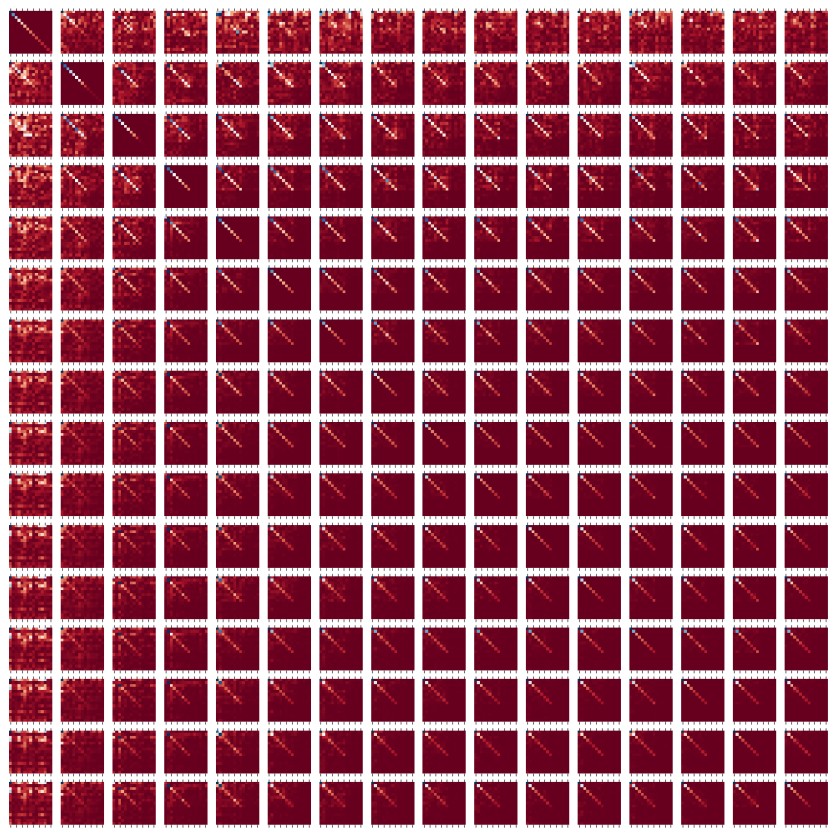

Figure 18: Fully-connected ResNet34 (Type 1 model) trained on FashionMNIST.

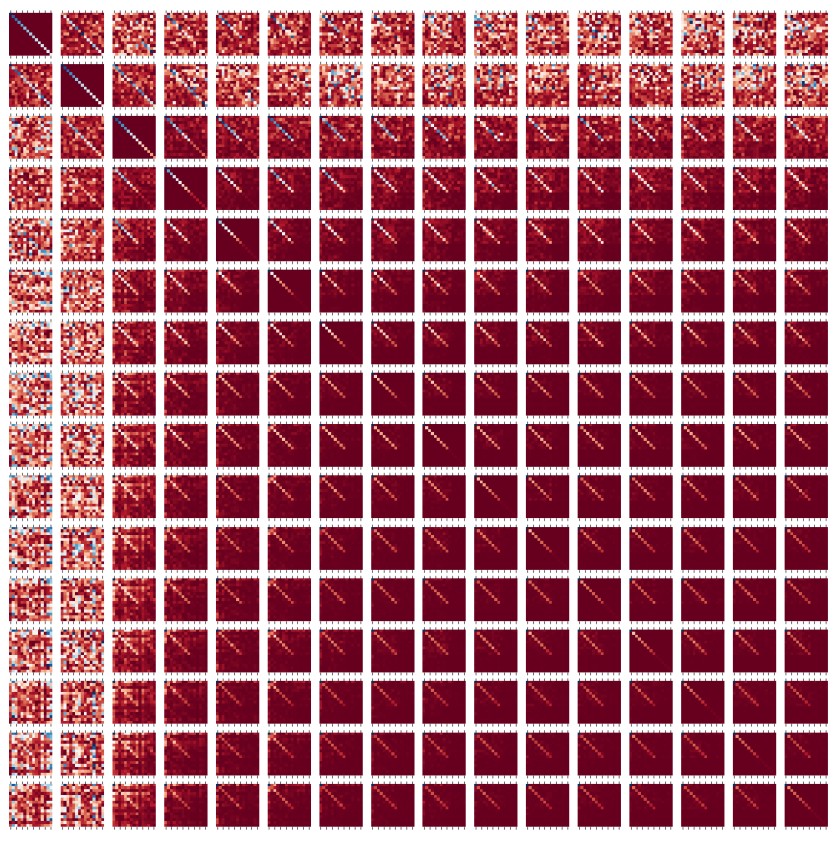

Figure 19: Fully-connected ResNet34 (Type 1 model) trained on CIFAR10.

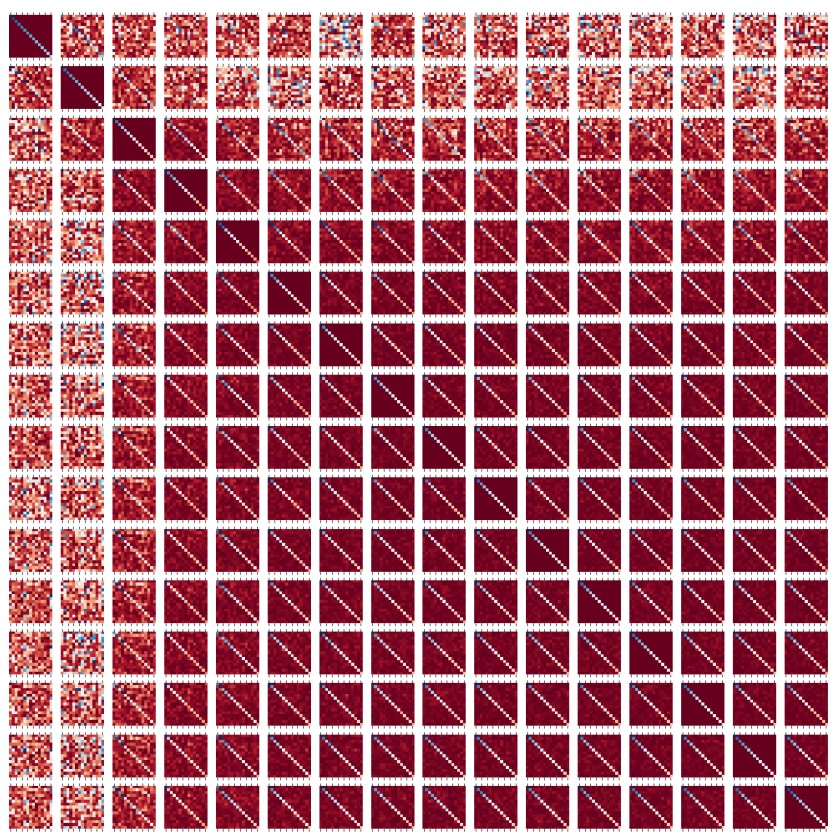

Figure 20: Fully-connected ResNet34 (Type 1 model) trained on CIFAR100.

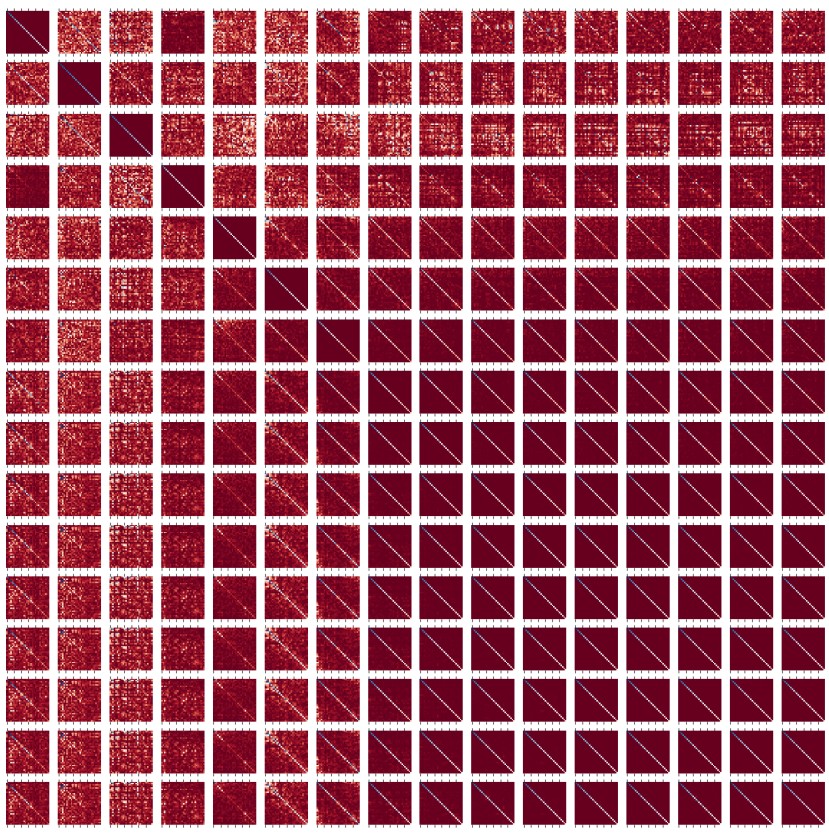

Figure 21: Convolutional ResNet34 (Type 2 model) trained on MNIST.

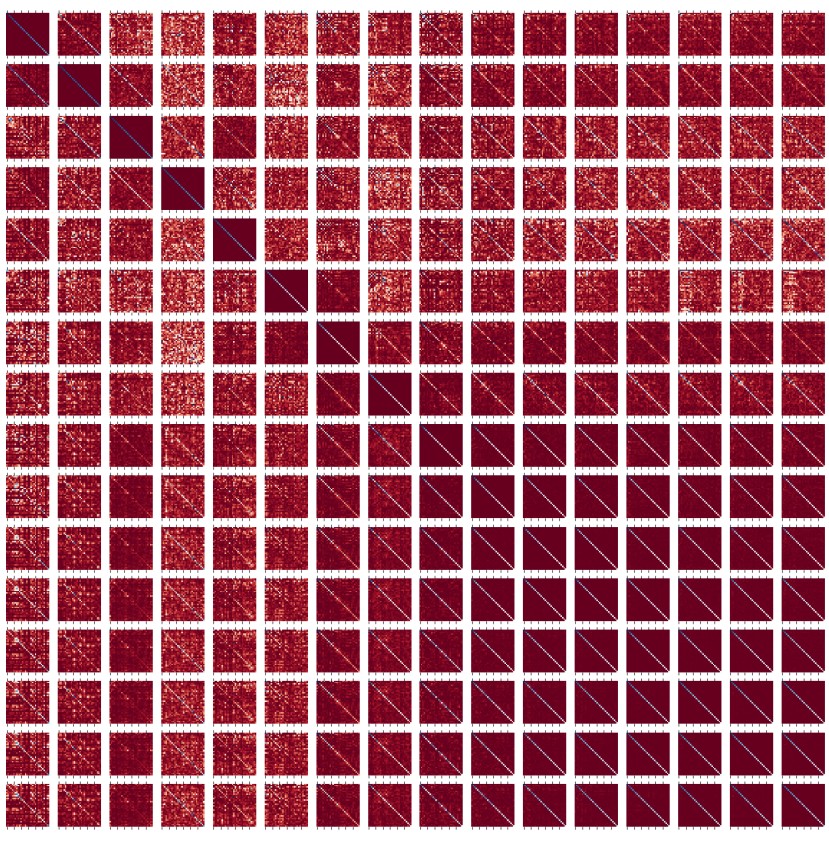

Figure 22: Convolutional ResNet34 (Type 2 model) trained on FashionMNIST.

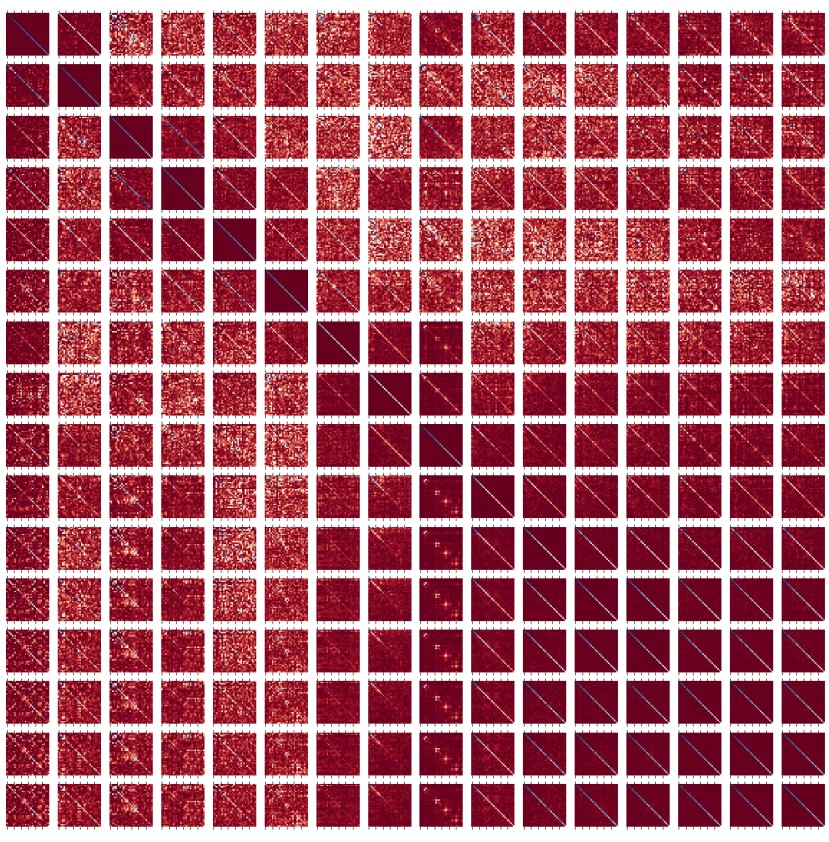

Figure 23: Convolutional ResNet34 (Type 2 model) trained on CIFAR10.

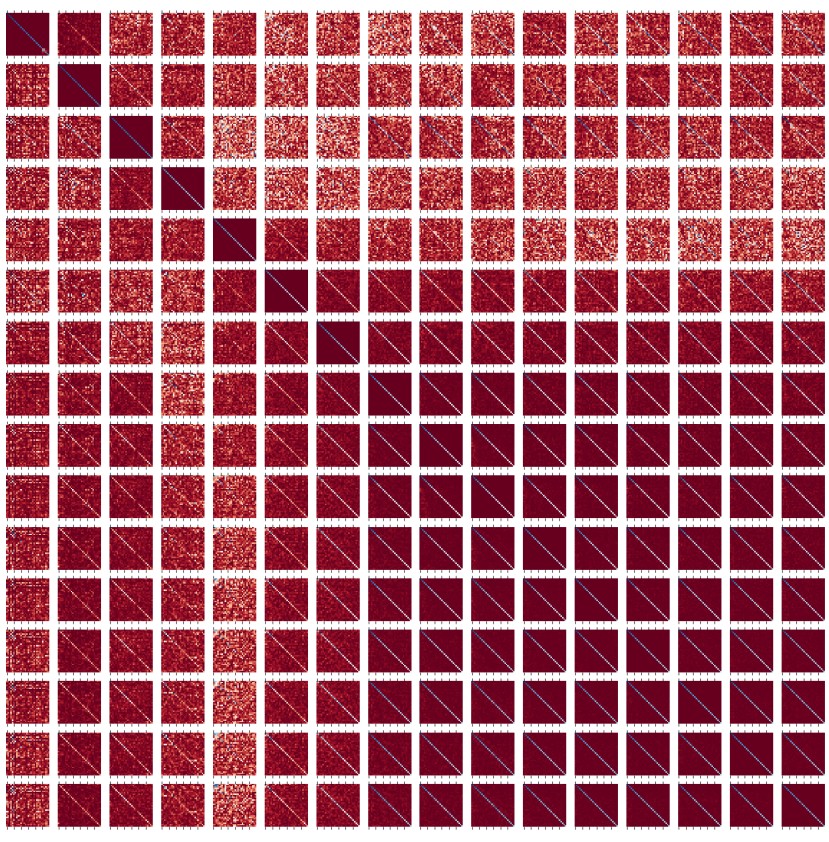

Figure 24: Convolutional ResNet34 (Type 2 model) trained on CIFAR100.

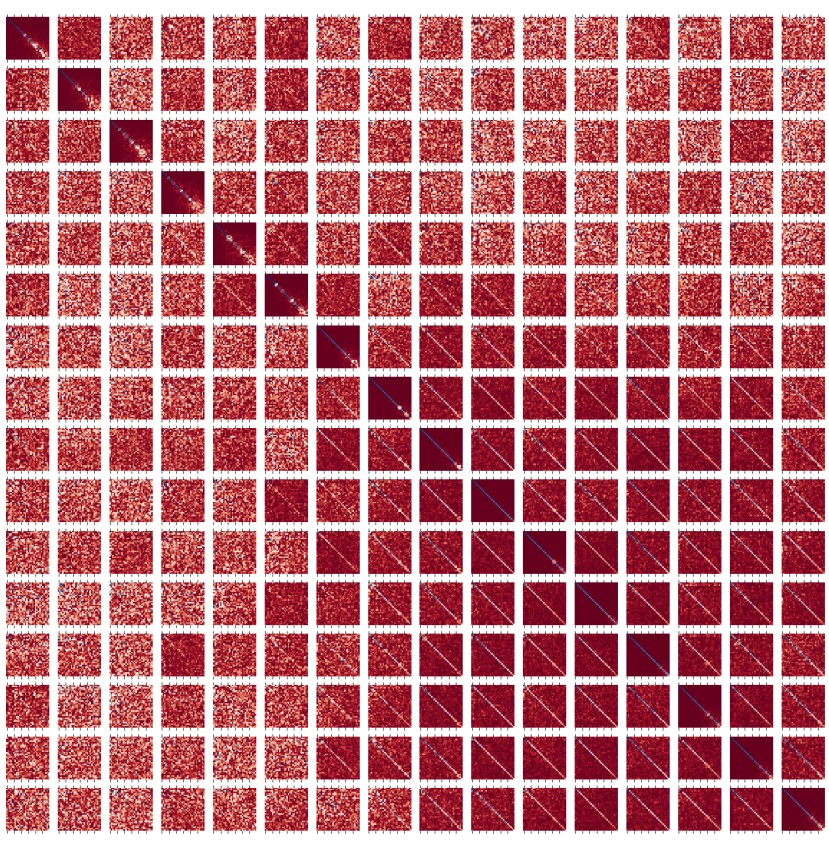

Figure 25: Convolutional ResNet34 (Type 2 model) trained on ImageNette.

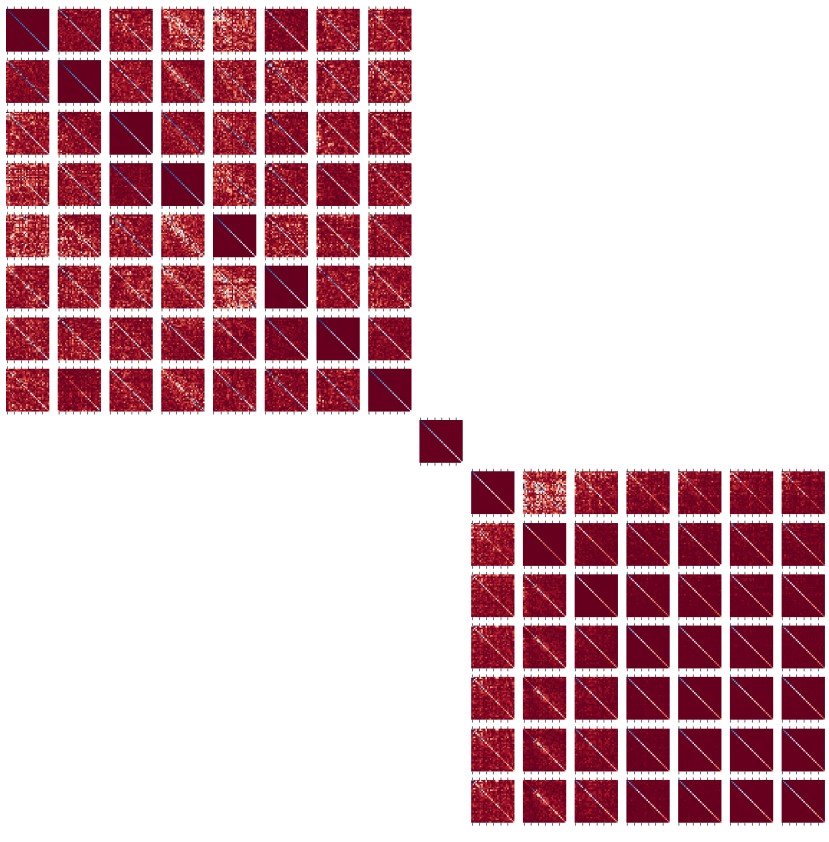

Figure 26: Convolutional ResNet34 with downsampling (Type 3 model) trained on MNIST.

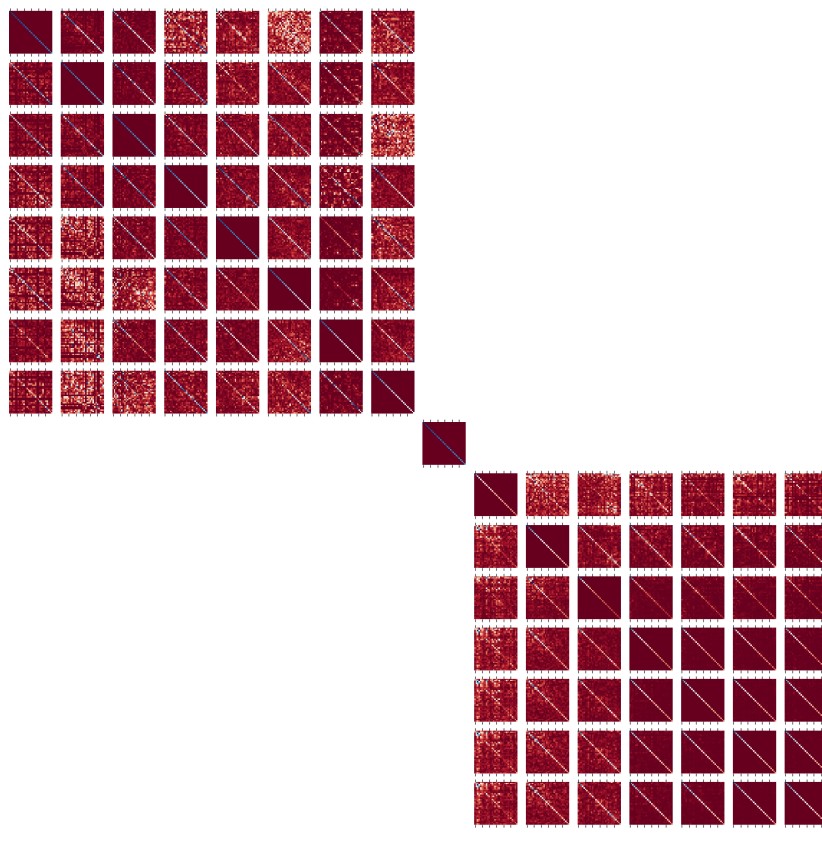

Figure 27: Convolutional ResNet34 with downsampling (Type 3 model) trained on FashionMNIST.

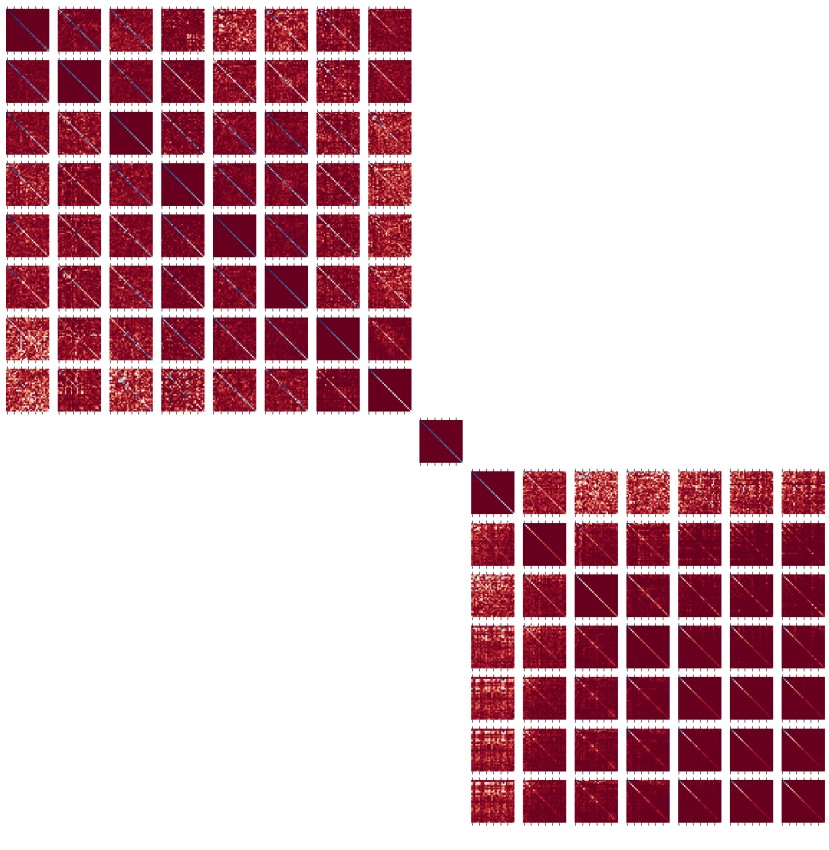

Figure 28: Convolutional ResNet34 with downsampling (Type 3 model) trained on CIFAR10.

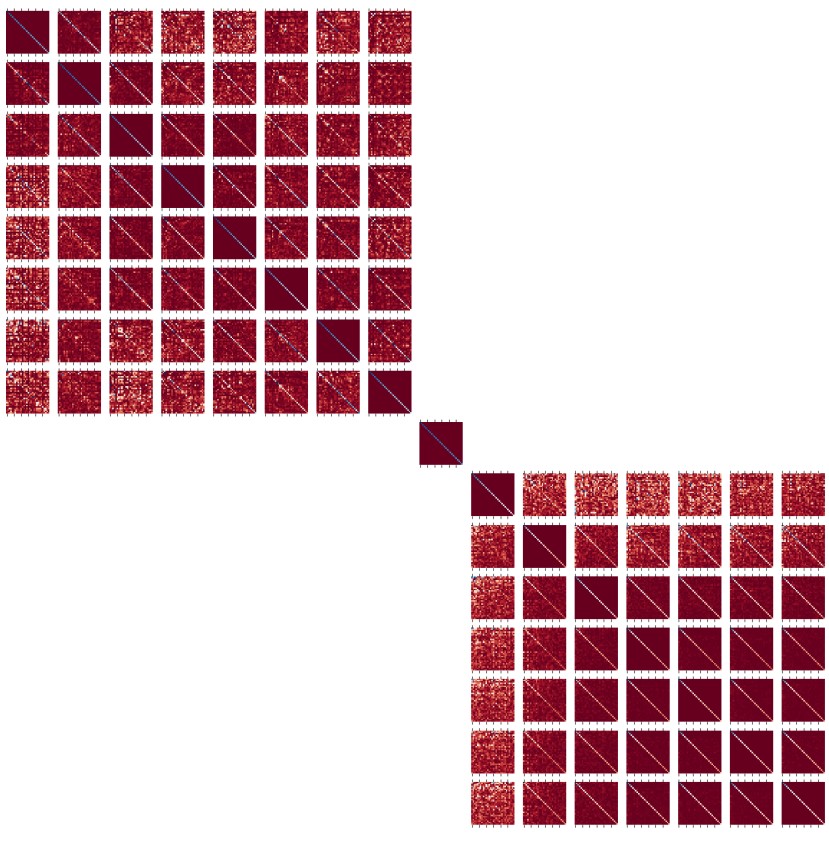

Figure 29: Convolutional ResNet34 with downsampling (Type 3 model) trained on CIFAR100.

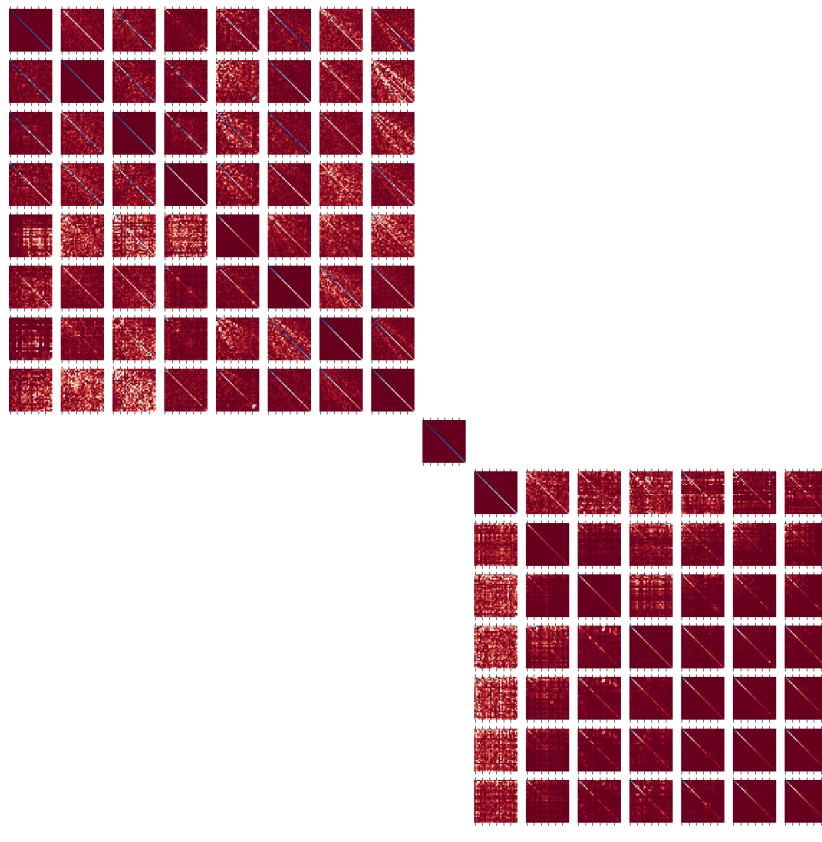

Figure 30: Convolutional ResNet34 with downsampling (Type 3 model) trained on ImageNette.

## C.3  (RA2): $V_{j,K}^{\top} J_i U_{j,K}$

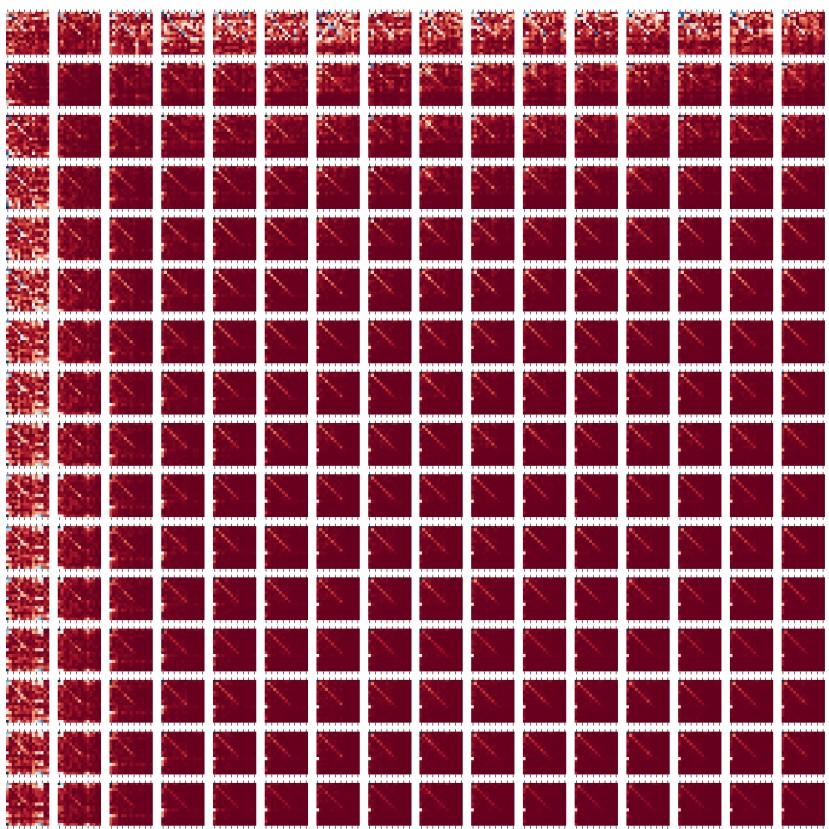

Figure 31: Fully-connected ResNet34 (Type 1 model) trained on MNIST.

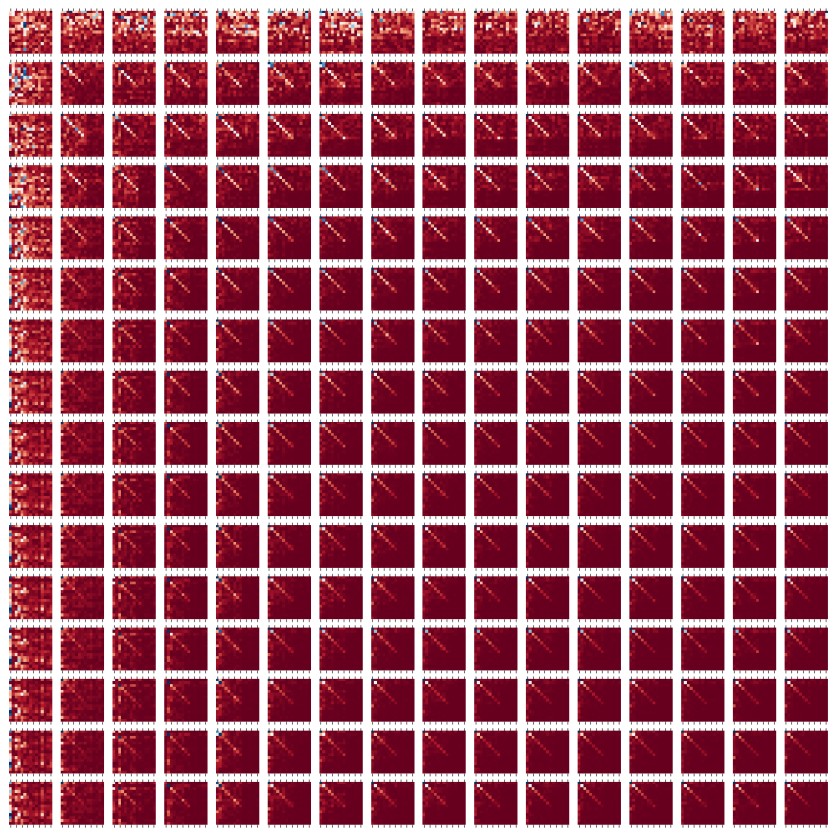

Figure 32: Fully-connected ResNet34 (Type 1 model) trained on FashionMNIST.

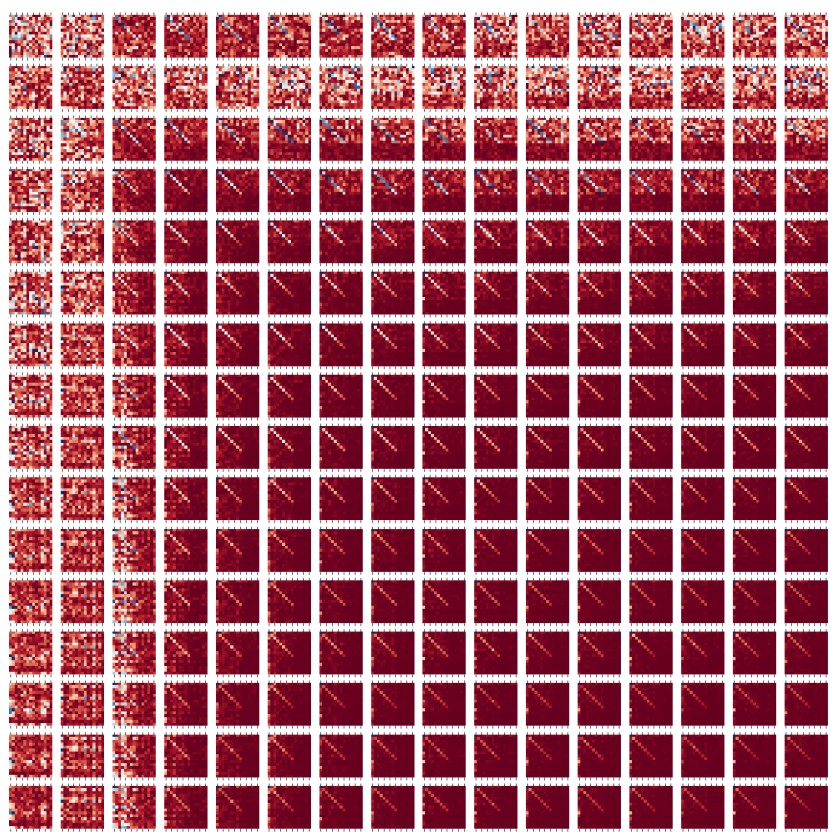

Figure 33: Fully-connected ResNet34 (Type 1 model) trained on CIFAR10.

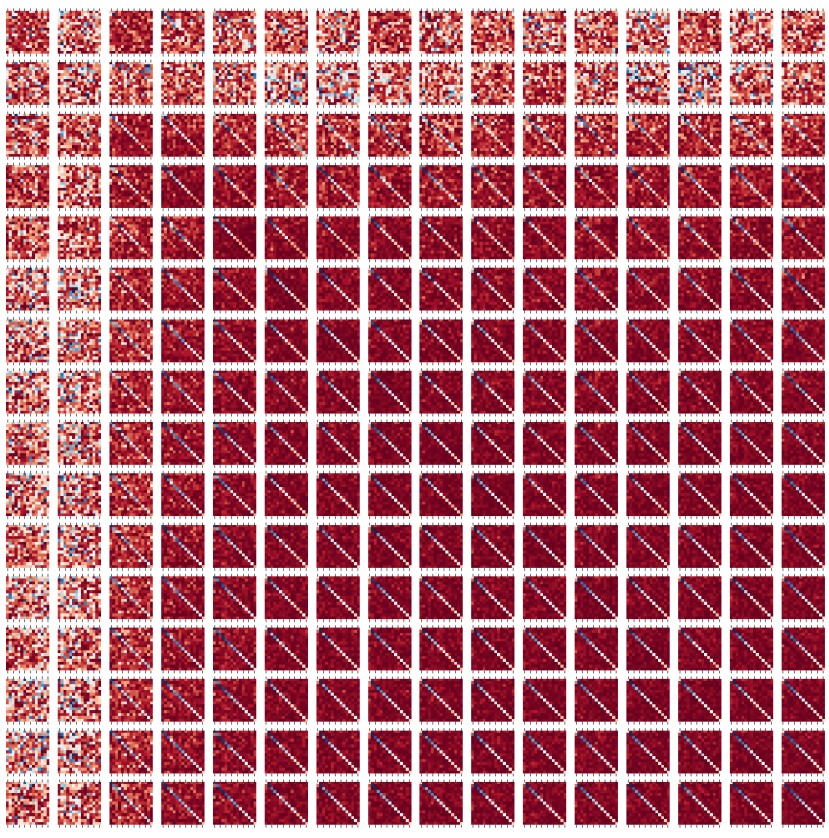

Figure 34: Fully-connected ResNet34 (Type 1 model) trained on CIFAR100.

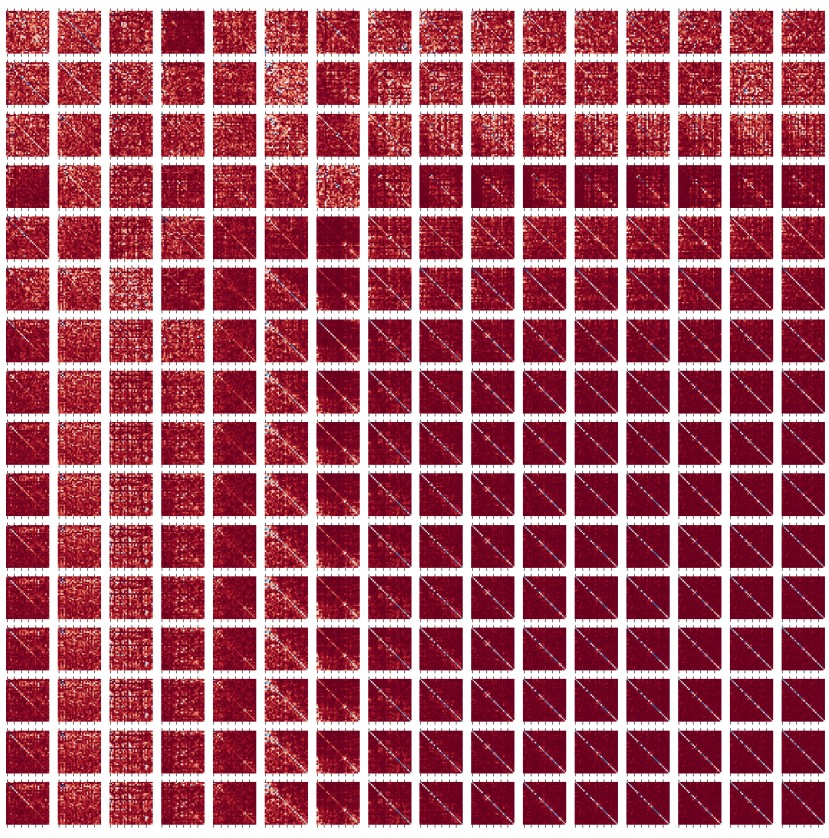

Figure 35: Convolutional ResNet34 (Type 2 model) trained on MNIST.

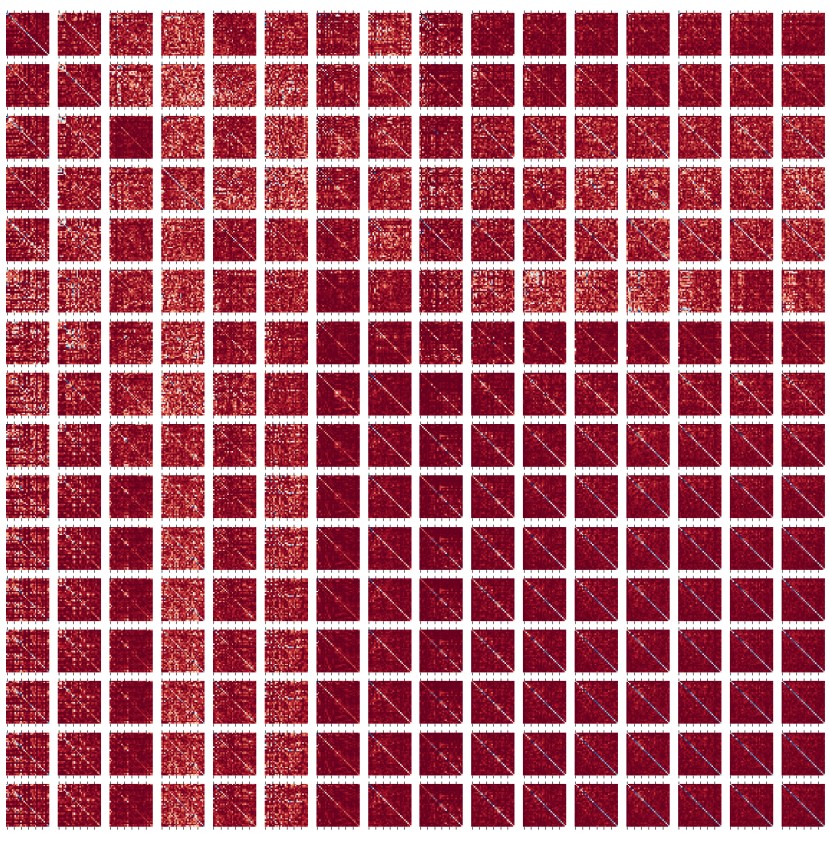

Figure 36: Convolutional ResNet34 (Type 2 model) trained on FashionMNIST.

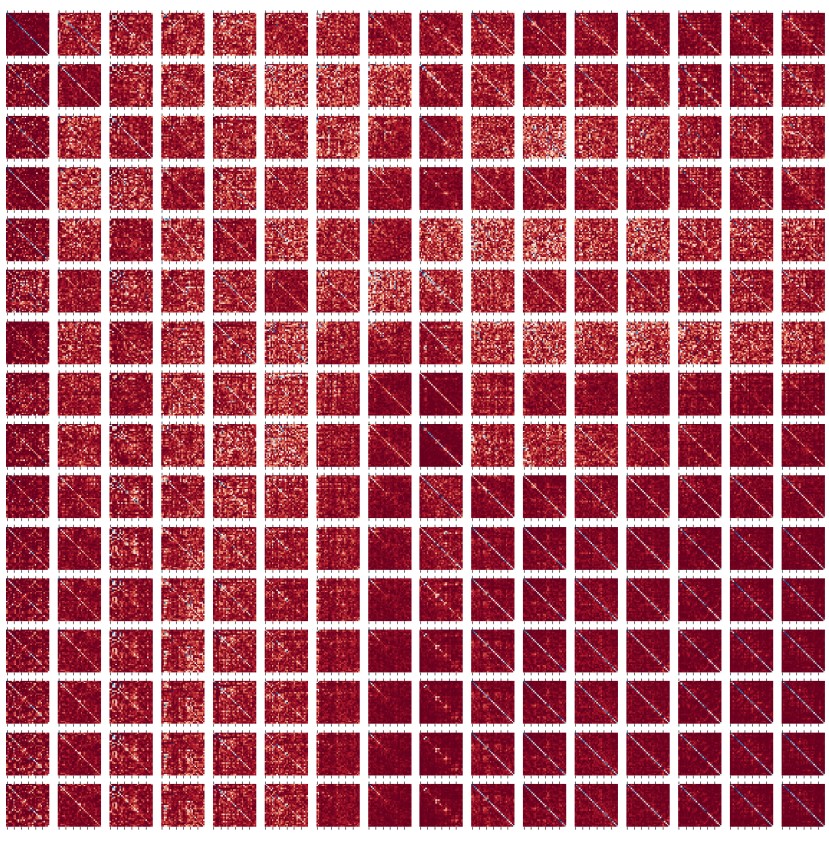

Figure 37: Convolutional ResNet34 (Type 2 model) trained on CIFAR10.

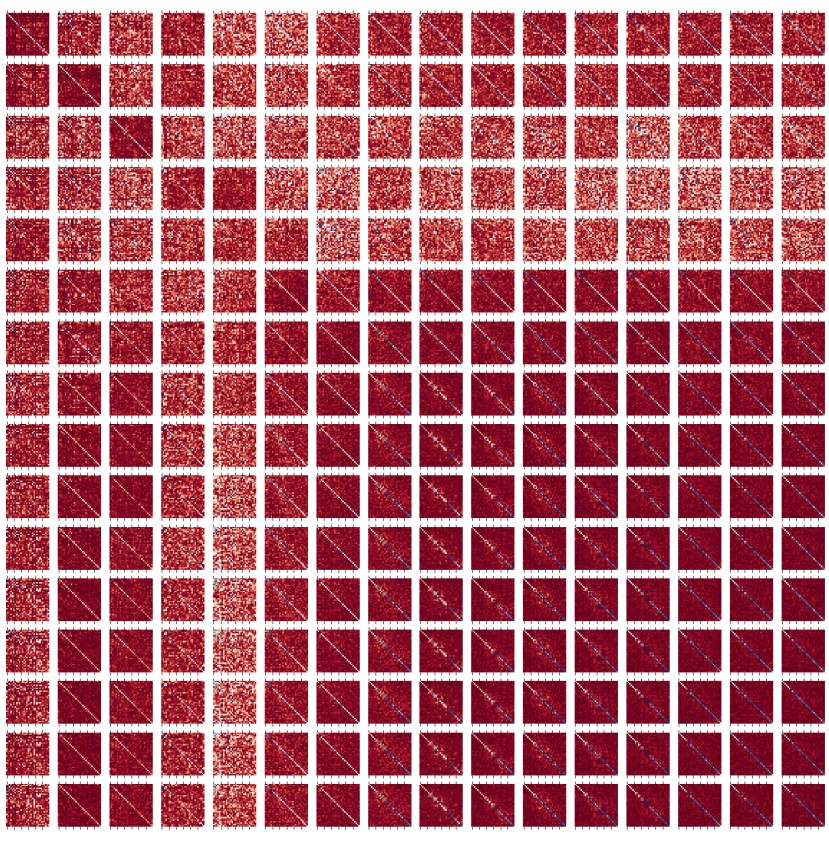

Figure 38: Convolutional ResNet34 (Type 2 model) trained on CIFAR100.

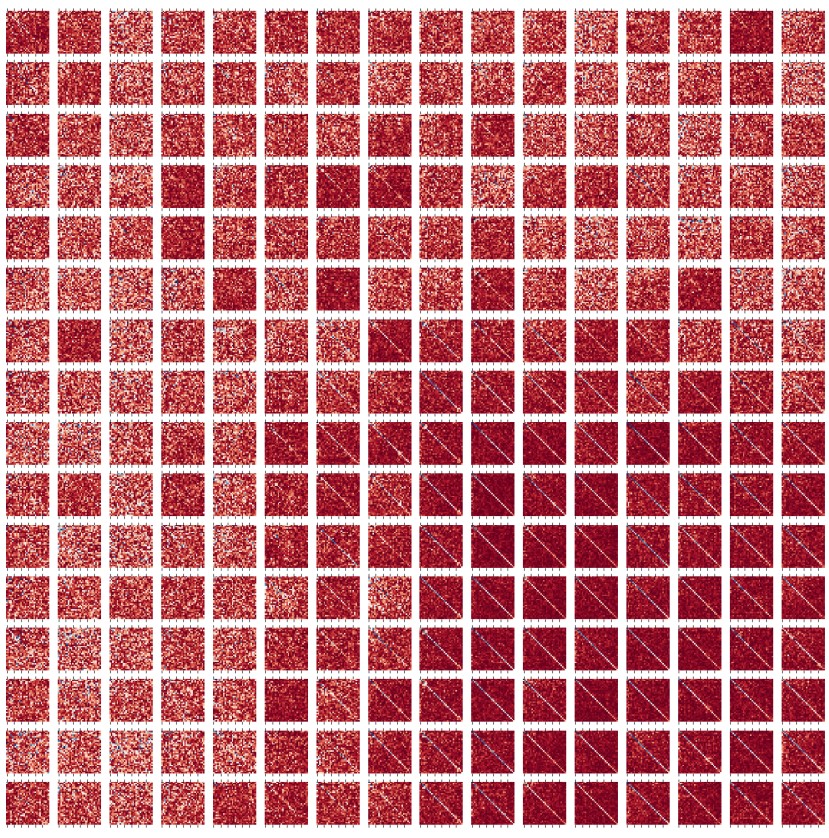

Figure 39: Convolutional ResNet34 (Type 2 model) trained on ImageNette.

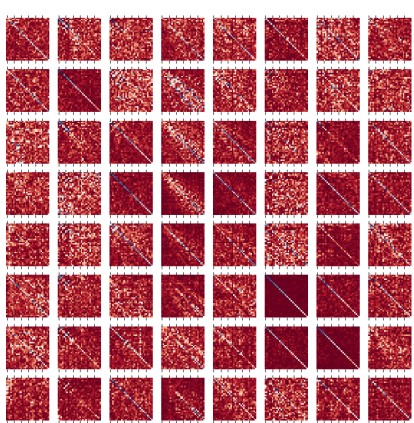

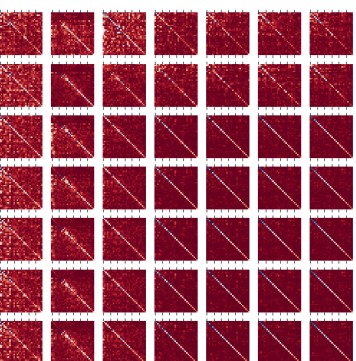

Figure 40: Convolutional ResNet34 with downsampling (Type 3 model) trained on MNIST.

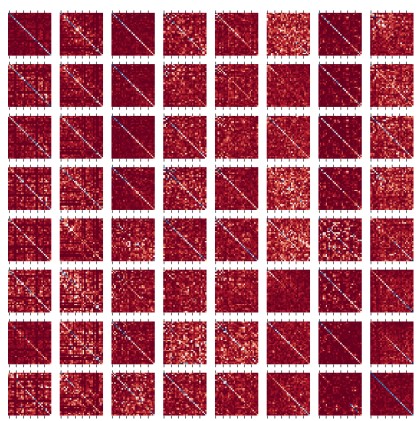

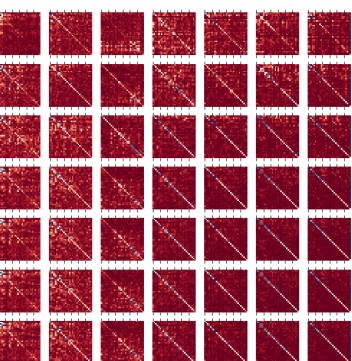

Figure 41: Convolutional ResNet34 with downsampling (Type 3 model) trained on FashionMNIST.

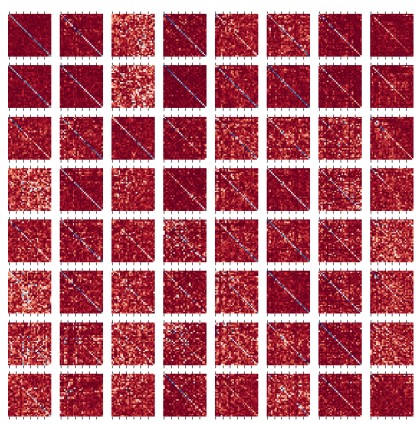

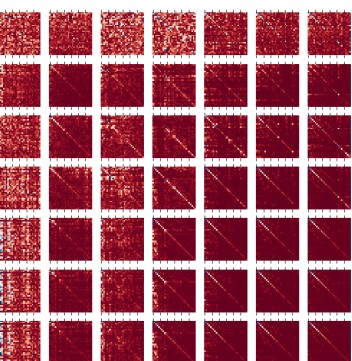

Figure 42: Convolutional ResNet34 with downsampling (Type 3 model) trained on CIFAR10.

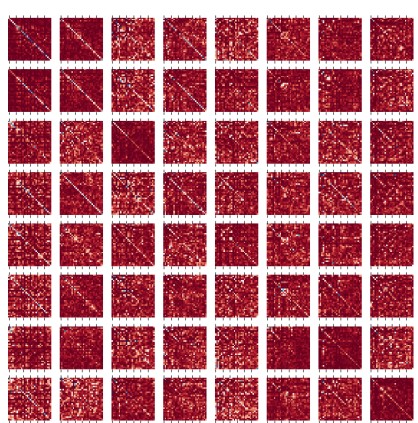

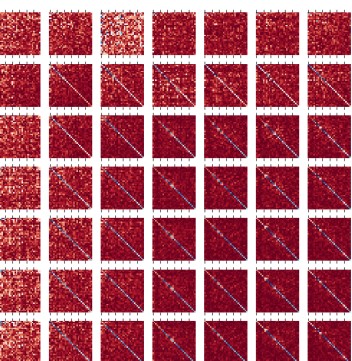

Figure 43: Convolutional ResNet34 with downsampling (Type 3 model) trained on CIFAR100.

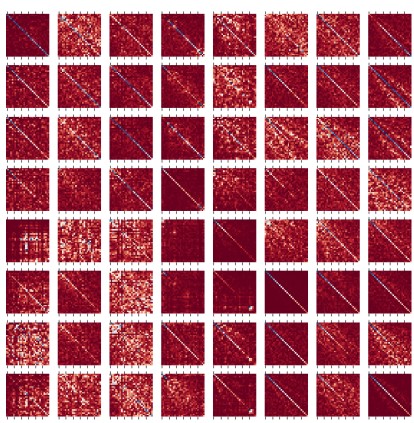

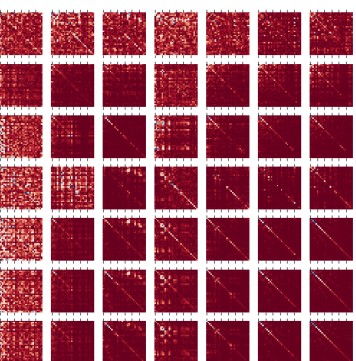

Figure 44: Convolutional ResNet34 with downsampling (Type 3 model) trained on ImageNette.

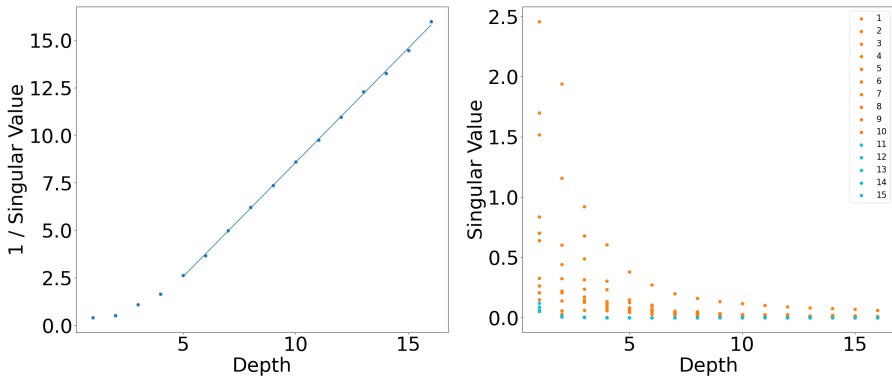

Figure 45: Fully-connected ResNet34 (Type 1 model) trained on MNIST.

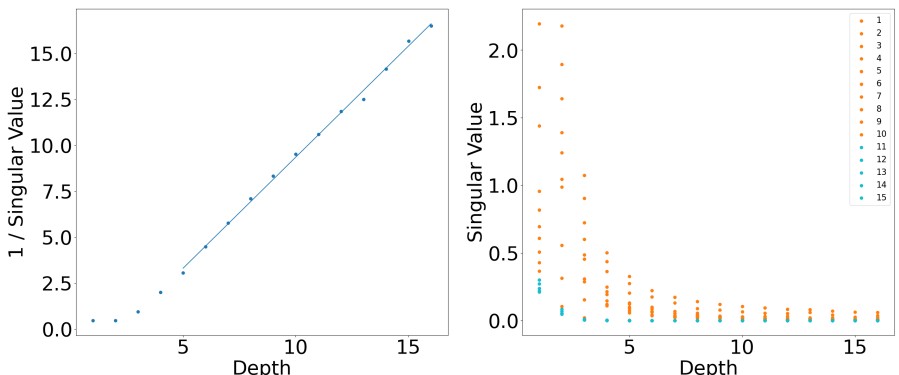

Figure 46: Fully-connected ResNet34 (Type 1 model) trained on FashionMNIST.

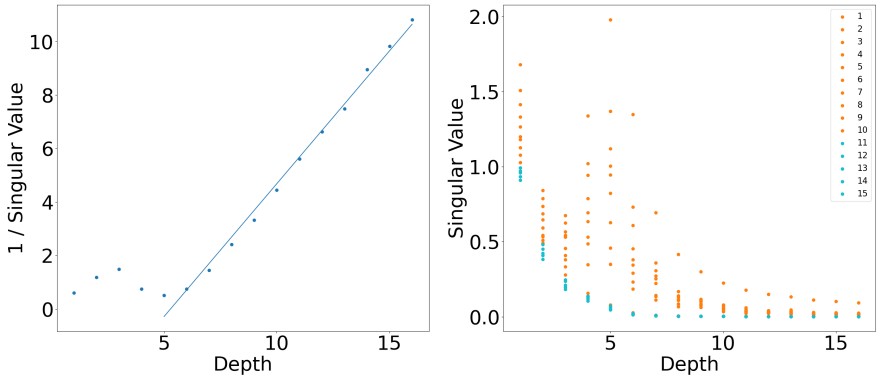

Figure 47: Fully-connected ResNet34 (Type 1 model) trained on CIFAR10.

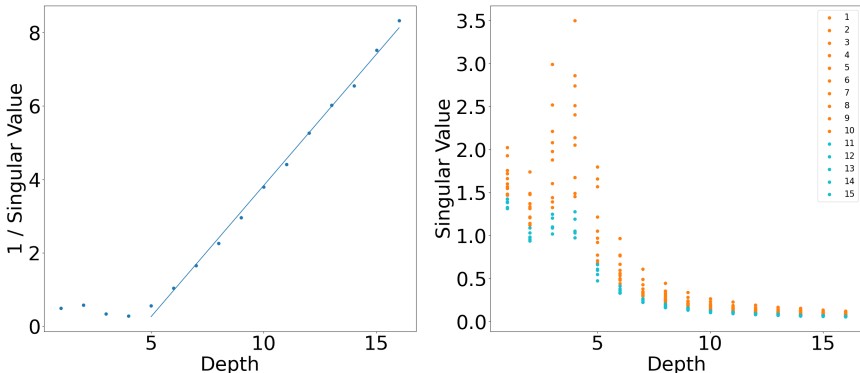

Figure 48: Fully-connected ResNet34 (Type 1 model) trained on CIFAR100.

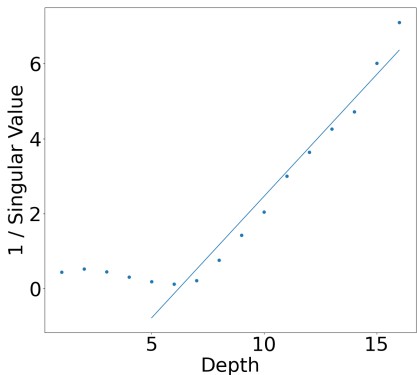

Figure 49: Convolutional ResNet34 (Type 2 model) trained on MNIST.

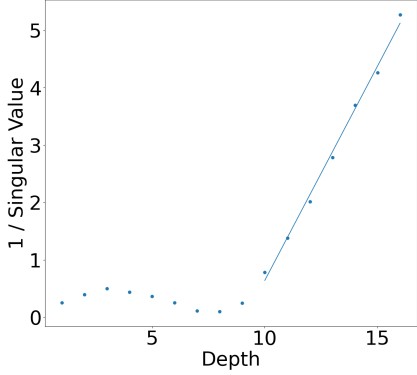

Figure 50: Convolutional ResNet34 (Type 2 model) trained on FashionMNIST.

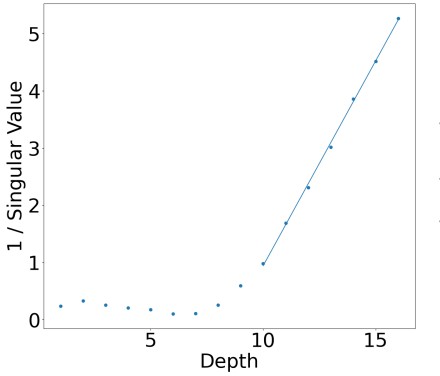

Figure 51: Convolutional ResNet34 (Type 2 model) trained on CIFAR10.

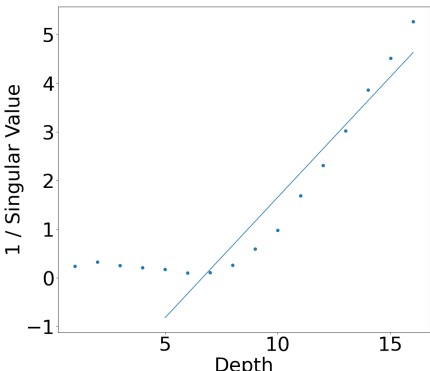

Figure 52: Convolutional ResNet34 (Type 2 model) trained on CIFAR100.

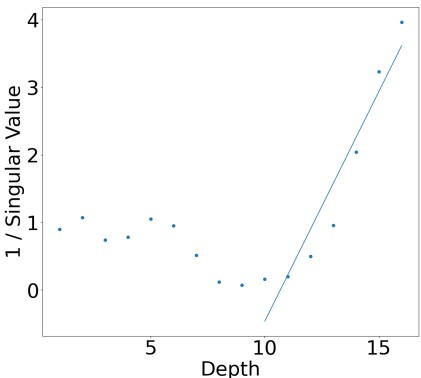

Figure 53: Convolutional ResNet34 (Type 2 model) trained on ImageNette.

## D Broader Impacts

This work presents foundational research and we do not foresee any potential negative societal impacts.

