# OpenReview forum: "Residual Alignment: Uncovering the Mechanisms of Residual Networks"
_NeurIPS.cc/2023/Conference — NeurIPS 2023 poster_

### Official Review · Reviewer_jJfi · 2023-07-04

**Soundness:** 4 excellent
**Presentation:** 4 excellent
**Contribution:** 4 excellent
**Rating:** 8
**Confidence:** 4

**Summary:**

This work discovers the phenomenon of Residual Alignment (RA) in ResNets, whereby the top left and right singular vectors of residual Jacobians align with each other and in between different residual blocks. Through extensive experimental verification as well as novel theoretical frameworks and derivations, the paper shows that RA naturally emerges in ResNets but does not emerge in non-residual networks. By directly linking RA to ResNets, this work sheds new light on the broad success of these architectures.

**Strengths:**

I believe this paper can have a significant impact.
- *Significance*: This work provides novel insights that have wide ramifications in a broad range of topics (deep learning theory, regularization, architecture design, model compression). Indeed, the precise mechanisms underpinning the success of ResNets remains a hot topic of research with high stakes. I believe this work makes a rare breakthrough in that direction.
- *Novelty*: To my knowledge, the discovery of RA and its precise mechanisms is novel. There is also novelty in both the experimental protocols and the theoretical frameworks. These protocols and frameworks might prove useful for the community.
- *Quality*: The analysis is highly convincing. I feel the experimental and theoretical evidences supporting the analysis are undebatable.
   - The experimental protocols are innovative and strongly convey the paper's claim, demonstrating both the presence of RA in residual networks and its absence in non-residual networks.
   - The theoretical analysis is equally innovative and excellent.
- *Contextualization*: The context of the work is appropriately provided (aside from minor points detailed below), with connections made to previous works, including to the latest developments on layer-wise Neural Collapse. The paper also details potential ramifications of the discovery of RA.
- *Clarity*: The paper is well-structured, concise, clear. The writing is good. The plots and figures are striking, providing the required evidence to support the claims.
- *Soundness*: The theoretical proofs are sound (aside from very minor points detailed below).
- *Reproducibility*: The authors released their code for reproducibility. The code is neat and clean, with high quality standards.

**Weaknesses:**

I see *no real weaknesses*, only minor points that can be easily addressed.

Minor points:
- Restriction to the context of classification. RA1 and RA3 are specific to classification, and all the tasks considered fall within such a context. If the findings are restricted to classification, I think this restriction should be clearly stated.
- I understand that the class-wise equi-spacing on a line of intermediate representations is a consequence of RA (in particular the scaling of top singular values inversely with depth), combined with the increased magnitude of $h_i$ due to the aggregated summation in ResNets. If this reasoning is correct, perhaps it would be worthwhile detailing it?
- The authors missed some previous works that studied ResNets vs non-residual networks at initialization, notably [1], [2] and [3].
- There seem to be typos on lines 160, 239, 243. Also shouldn't "these values" be replaced by "the top-1 singular values" in the caption of Figure 5?
- It seems Theorem 3.1 is only proved in the context of binary classification. Perhaps this should be stated. Also, I believe there is a term missing in the Equation following line 11 in the Appendix. That term relates to $h_1 - U_{k,1} U_{k,1}^T h_1$ (as can be seen e.g. in the case where all $S_i$ would be equal to zero).
- Perhaps Theorem 2 in the Appendix should state the convention of positive singular values, meaning that inequality on the $\sup$ of the trace would always be guaranteed, with equality obtained if the determinant's sign equals 1. Also shouldn't $L+1$ be replaced by $L$ on line 68?

References:

[1] The Shattered Gradients Problem: If ResNets are the answer, then what is the question?, D. Balduzzi et al., ICML 2017

[2] Gradients Explode - Deep Networks Are Shallow - ResNet Explained , G. Philipp et al., ICLR Workshop 2018

[3] Characterizing Well-Behaved vs. Pathological Deep Neural Networks, A. Labatie, ICML 2019

**Questions:**

I would be keen to have the authors' opinion on the following points:
- What happens outside the context of classification (e.g. in regression, object detection, etc)?
- Could you expand on the connection between RA and layer-wise Neural Collapse? It seems layer-wise Neural Collapse was originally found to be present in both ResNets and non-residual networks (notably VGGs). Given your findings concerning the absence of RA in non-residual networks, does that mean layer-wise Neural Collapse can occur without RA?
- Could you confirm that RA is still in agreement with the experimental findings in Veit et al., 2016? If so, should Veit et al., 2016 be reinterpreted with a new light?
- Could you express your view on the connection between RA and stochastic depth? If all residual blocks have aligned singular vectors, shouldn't the effect of stochastic depth be a plain random multiplicative factor of the logits?

**Limitations:**

If the authors' findings are restricted to the context of classification, perhaps this restriction should be more clearly stressed.

---

> ### Author Rebuttal · Authors · 2023-08-10
>
> We are grateful to the reviewer for the time and effort they must have dedicated to providing such a thorough and constructive review. Below, we respond to each point that was raised.
>
> > RA1 and RA3 are specific to classification, and all the tasks considered fall within such a context. If the findings are restricted to classification, I think this restriction should be clearly stated. What happens outside the context of classification (e.g. in regression, object detection, etc)?
>
> Indeed, our measurements of RA are restricted to classification problems. Since we can not presume RA extends to other settings, we will explicitly mention in the manuscript that we are considering multiclass classification problems.
>
> At this point, we have no intuition as to whether RA would occur for regression or object detection. It would be very interesting to explore the question empirically or even theoretically in future work.
>
>
> > I understand that the class-wise equi-spacing on a line of intermediate representations is a consequence of RA (in particular the scaling of top singular values inversely with depth), combined with the increased magnitude of h_i due to the aggregated summation in ResNets. If this reasoning is correct, perhaps it would be worthwhile detailing it?
>
> The reasoning is precisely correct! In fact, this is a crucial step in the proof of Theorem 3.1 (second to last equation on page 1 of the Appendix). Following your suggestion, we will move this portion of the proof into the manuscript so as to provide a stronger intuition regarding RA.
>
>
> > The authors missed some previous works that studied ResNets vs non-residual networks at initialization.
>
> Thank you. We now cite these relevant works.
>
>
> > There seem to be typos on lines 160, 239, 243. Also shouldn't "these values" be replaced by "the top-1 singular values" in the caption of Figure 5?
>
> Thank you. We have fixed these issues.
>
>
> > It seems Theorem 3.1 is only proved in the context of binary classification. Perhaps this should be stated.
>
> We apologize for this omission. We now explicitly state this in the Theorem.
>
>
> > Also, I believe there is a term missing in the Equation following line 11 in the Appendix. That term relates to $h_1 - U_{k,1} U_{k,1}^T h_1$.
>
> Thank you for catching this mistake. We corrected the proof by adding the missing term. The final conclusion from the theorem does not change as the intermediate representations are still situated on a line, except the line is now shifted by a constant vector.
>
>
> > Perhaps Theorem 2 in the Appendix should state the convention of positive singular values.
>
> Thanks for highlighting this. We will add to Theorem 2 the convention of positive singular values.
>
>
> > Also shouldn't L+1 be replaced by L on line 68?
>
> Thank you for pointing out this mistake.
>
>
> > Could you expand on the connection between RA and layer-wise Neural Collapse? It seems layer-wise Neural Collapse was originally found to be present in both ResNets and non-residual networks (notably VGGs). Given your findings concerning the absence of RA in non-residual networks, does that mean layer-wise Neural Collapse can occur without RA?
>
> A fascinating question. Indeed, current empirical evidence suggests that layer-wise Neural Collapse can occur without RA. From a different perspective, RA can be thought of as a more structured or regularized version of layer-wise Neural Collapse where Residual Jacobians are forced to align. At this point, it is unclear what benefits such structure or regularization confers to generalization or robustness. We hope future work investigates this intriguing question.
>
> On a related note, the enclosed PDF presents an experiment demonstrating the co-occurrence of Neural Collapse and Residual Alignment.
>
>
> > Could you confirm that RA is still in agreement with the experimental findings in Veit et al., 2016? If so, should Veit et al., 2016 be reinterpreted with a new light?
>
> RA is indeed in agreement with Veit et al. Consider a simple linear ODE:
> $$x' = Ax.$$
>
> Its solution is given by
> $$x(t) = \exp(t A) x(0).$$
>
> Recalling the following identity
> $$\exp(y) = \lim_{L \to \infty} (1+y/L)^L,$$
>
> we can write
> $$x(t) = \lim_{L \to \infty} (I + t A/L)^L x(0),$$
>
> or equally
> $$x(t) = \lim_{L \to \infty} (I + t A/L) (I + t A/L) \dots (I + t A/L) x(0),$$
>
> where the product is repeated $L$ times. Expanding the brackets shows that $x(t)$ can be thought of as a collection of many paths of various lengths, due to the binomial identity.
>
> Veit et al. state:
> "...residual networks can be viewed as a collection of many paths."
> Ignoring constants, they view ResNets as
> $$x(t) = (I + t A_1/L) (I + t A_2/L) \dots (I + t A_L/L) x(0).$$
>
> Notice how they do not make any claims about the alignment of the various $A_i$ matrices. We view ResNets as implementing
> $$x(t) = (I + t A/L) (I + t A/L) \dots (I + t A/L) x(0),$$
>
> where all the $A_i$ matrices are aligned. One benefit of this interpretation is that, as $L \to \infty$, we can write $x(t)$ in closed form, namely $x(t) = \exp(t A) x(0)$.
>
> The above intuition is imprecise since the singular values should scale inversely proportionally with depth. Still, we believe it explains the difference in perspective between Veit et al. and our work.
>
>
> > Could you express your view on the connection between RA and stochastic depth? If all residual blocks have aligned singular vectors, shouldn't the effect of stochastic depth be a plain random multiplicative factor of the logits?
>
> We agree. However, in practice, RA falls short of perfection. The enclosed PDF presents an experiment demonstrating that introducing stochastic depth enhances RA even further.
>
> >
>
> In closing, we'd like to express again our gratitude to the reviewer for their feedback which not only improved the current state of our manuscript but also set the stage for intriguing future investigations. In light of these refinements, should you find it fitting, we would be most grateful for any potential increase in our score.

---

> > ### Comment · Reviewer_jJfi · 2023-08-12
> > **Rebuttal update**
> >
> > I thank the authors for their detailed response, which I highly appreciated.
> >
> > The additional experiments conducted by the authors are enlightening. Specifically, the fact that stochastic depth reinforces RA by discouraging heterogeneous residual Jacobians $J_i$ is compelling.
> >
> > As pointed out by another reviewer, it might still be advantageous to clarify the dependence of Jacobians on inputs:
> >
> > - If that dependence is overlooked, does it mean that input dependence is assumed to be weak?
> >
> > - In the Figures that explore the dependence between the rank of Jacobians and the number of classes (i.e. outlining a tie between Jacobians and data distribution), are Jacobians computed for a single input, or are they averaged over the whole data distribution?
> >
> > Overall, I would like to keep my rating and remain highly positive.

---

> > > ### Author Response · Authors · 2023-08-12
> > >
> > > We would like to thank the reviewer for their additional feedback. For every figure, we compute the Jacobians using a single randomly selected training example. While different input examples can cause variations in the figures, the influence of the input is weak, as mentioned by the reviewer. We will include a figure in the Appendix to demonstrate the variations in RA properties due to varying inputs.

---

> > > > ### Comment · Reviewer_jJfi · 2023-08-13
> > > >
> > > > Thank you very much for your prompt reply.
> > > >
> > > > Everything is clear and highly positive on my end.

---

### Official Review · Reviewer_Tohn · 2023-07-06

**Soundness:** 4 excellent
**Presentation:** 1 poor
**Contribution:** 4 excellent
**Rating:** 7
**Confidence:** 3

**Summary:**

The paper "Residual Alignment: Uncovering the Mechanisms of Residual Networks" explores the underlying mechanisms and success factors of the ResNet architecture, which has gained significant popularity in deep learning. The authors conduct an empirical study by linearizing the residual blocks of ResNet using Residual Jacobians and measuring their singular value decompositions.

They introduce the concept of Residual Alignment (RA) characterized by four properties: equispaced intermediate representations, alignment of singular vectors, low rank of Residual Jacobians, and inverse scaling of singular values. The phenomenon of RA is observed in well-generalizing models and is absent when skip connections are removed. The authors also propose a mathematical model that demonstrates the occurrence of RA.

*Contribution & significance:* The paper identifies an interesting and seemingly novel phenomenon which they term residual alignment. They show that this observation is on solid theoretical grounds by studying the binary classification problem with cross entropy loss.  This sheds new light on an the role of residual connections in neural nets and will contribute to our overall understanding of this importance architectural component.

*Major writing issues* The main drawback of this paper is its writing quality which is very low at the moment. While I grasped the main ideas by going through the appendix, several of critical pieces of information, such as definition of residual alignment Jacobians ($J_i$'s), are missing from the main text. This will make it very difficult to understand this paper just by reading the main text (which should be the default assumption for a reader). This is quite unfortunate given that the contributions are technically strong & very interesting. Thus, a significant overhaul of the writing seems to be necessary to make this work publishable.

*Clarity about novelty* It would be helpful if authors clarify the new insights and novelty of this work, in contrast to the paper that they numerously cite "A Mathematical Principle of Deep Learning: Learn the Geodesic Curve in the Wasserstein Space". Also adding more related literature will also be helpful to the readers.

**Strengths:**

- Identification of Residual Alignment: The paper identifies and characterizes an interesting and seemingly novel phenomenon of Residual Alignment (RA), and highlights its four key properties. The properties of RA are logically interrelated and consistently observed across various architectures. The authors present evidence to support the existence of RA.

- The authors conducted a comprehensive empirical study of the ResNet architecture, linearizing residual blocks using Residual Jacobians and analyzing their singular value decompositions. This approach provides valuable insights into the underlying mechanisms of ResNet models.

- Theoretical Proofs and Abstraction: One of the strongest points of this paper seems to be the theoretical proofs for the emergence of RA in the setting of binary classification with cross-entropy loss. The introduction of the Unconstrained Jacobians Model as a mathematical abstraction adds further depth to the analysis and strengthens the paper's theoretical foundations.

- The proof the main theorem in the appendix is interesting in its own right. They authors first linearize the loss function and then approximate it as a product of $(I+J_i)$ terms where $J_i$'s are the Jacobians. They go on to argue that the loss of a general configurtion (non-aligned singular vectors) is surprisingly bounded by a term that is equivalent to the aligned Jacobians. This is mainly done by invoking an interesting mathematical theorem. While I did not go into full detail of the proof and there might be details/flaws that I missed, the overall proof strategy makes sense and is very interesting, as it breaks down a very intricate problem to a tractable one. (As a side point, I recommend authors to conduct a few more detailed experimentation on the details of these steps, namely by directly. testing the inequalities they arrive at).


**Weaknesses:**


# Major issues with writing & presentation of results

One of the main drawbacks of the current manuscript is its presentation. There are major problems with the flow and writing which hinder my understanding of some details. Another major issues is the lack of enough related works and connection to the existing machine learning & statistics literature. Overall, I would recommend authors to do a major revision of this manuscript to make it publishable and readable.

 I tried to compile a list of minor issues with the writing to help with this.

- Fig 1 caption: “s true label and connected to form a trajectory” every connected line goes from input to output? Shouldn’t it have 34 layers (it looks like fewer dots than 34)

- Equation line 34-35: what is $I$?
- Equation between 57 & 58, what’s the dimensionality of $ \sigma’(…) $ in this equation?
- Non capitalised sentences (examples 68, 70, 74, 92, 93, several more )
- Jacobians are input-dependent, how are the Figures 2 & 3 made? Do the figures correspond to a single sample input? or averaged across multiple inputs?
- What does index $i$ in $J_i$ stand for?  This seems to be defined in the appendix but in the main text?
- line 59 “excluding the contribution from the skip connection.” What does this mean?



**Questions:**

One of my main questions concerns the background of this work. The relevant (recent) literature seems to rather scarce. The only directly related literature currently is "A Mathematical Principle of Deep Learning: Learn the Geodesic Curve in the Wasserstein Space". So my question is two folds:
- Do the authors believe there is no other related literature on this phenomenon of "emerging singular vector alignment"? Given the importance of this topic and its relevant to modern models, it seems to me that there should be more "adjacent" studies on the topic.
- Given that the aforementioned paper "A mathematical ... " does have quite a bit of overlap with the present work, can the authors make it clear how are their contributions different from this paper? In particular, the residual alignment phenomenon has been extensively discussed in the other paper with largely overlapping settings. It would be helpful for readers to understand what is the additional insights, ideas, proofs, etc, in this present paper.

**Limitations:**

yes

---

> ### Author Rebuttal · Authors · 2023-08-10
>
> We would like to extend our appreciation to the reviewer for their diligent and thorough evaluation of our paper. In what follows, we address carefully each raised point.
>
>
> > Fig 1 caption: “s true label and connected to form a trajectory” every connected line goes from input to output? Shouldn’t it have 34 layers (it looks like fewer dots than 34)
>
> Thank you for this question. Indeed, every connected line goes from the input to the output. Following the convention set by the original ResNet paper, a ResNet34 has $(34-2)/2=16$ residual blocks. The reason for this is that each residual block consists of two weight matrices and two layers are subtracted—one due to the initial convolution and the other at the final classification layer. Therefore, we would expect to see at most $16$ dots instead of $34$. In the plot, we can count around $10$ dots but the rest are too cluttered around the center. This is in agreement with Figure 40 in the Appendix showing the inverse scaling of the top singular value occurs for roughly the last $10$ blocks, which correspond to the $10$ visible dots. We will add a footnote to the manuscript explaining the above.
>
>
> > Equation line 34-35: what is I?
>
> We apologize for this oversight. In the revised manuscript, we clarify that the function I denotes the Initial layers (comprising a sequence of convolution, batch normalization, and ReLU operations) that occur prior to the first residual block within a ResNet architecture.
>
>
> > Equation between 57 & 58, what’s the dimensionality of in this equation?
>
> Thank you for this question. In the revised manuscript we clarified that the Residual Jacobians are square matrices of size $d \times d$, where $d$ is the number of elements in a ResNet intermediate representation. For instance, for a type 1 model, $d=512$ where $512$ is the width of the network. For a type 2 model, $d=64 \cdot 16 \cdot16=16384$ because the representations have $64$ channels and a spatial dimension resolution of $16 \times 16$.
>
>
> > Non capitalised sentences (examples 68, 70, 74, 92, 93, several more )
>
> We apologize for the oversight. We have rectified these errors, and all the sentences are now correctly capitalized.
>
>
> > Jacobians are input-dependent, how are the Figures 2 & 3 made? Do the figures correspond to a single sample input? or averaged across multiple inputs?
>
> The Figures correspond to a single randomly sampled input. We have incorporated this information into the Figures' caption.
>
>
> > What does index $i$ in $J_i$ stand for? This seems to be defined in the appendix but in the main text?
>
> We apologize for this omission. The index $i$ stands for the index of the residual block, i.e., its depth. We will clarify this in the main text.
>
>
> > line 59 “excluding the contribution from the skip connection.” What does this mean?
>
> Following Equation 35, recall a residual block evaluates
> $$h_{i+1} = \sigma (h_i + \mathcal{F} (h_i; \mathcal{W}_i)).$$
>
> Notice that the derivative of the residual block with respect to its input is given by
> $$\sigma’(h_i + \mathcal{F} (h_i; \mathcal{W}_i)) \left( I + \frac{\partial \mathcal{F} (h_i; \mathcal{W}_i)}{\partial h_i} \right),$$ where $I$ is an identity matrix.
>
> In line 59, we exclude the contribution from the skip connection, i.e., the term
> $$\sigma’(h_i + \mathcal{F} (h_i; \mathcal{W}_i)) I.$$
>
> We have added the above clarification to the revised manuscript.
>
>
> > Do the authors believe there is no other related literature on this phenomenon of "emerging singular vector alignment"? Given the importance of this topic and its relevant to modern models, it seems to me that there should be more "adjacent" studies on the topic.
>
> We appreciate the reviewer's recognition of the importance and relevance of our work in the context of modern models. In Section 5, we diligently acknowledge and reference all empirical and theoretical research that explores the fundamental mechanisms contributing to the remarkable success of ResNets. As far as we are aware, no one has previously mentioned the alignment of singular vectors of linearized residual blocks.
>
>
> > Given that the aforementioned paper "A mathematical ... " does have quite a bit of overlap with the present work, can the authors make it clear how are their contributions different from this paper? In particular, the residual alignment phenomenon has been extensively discussed in the other paper with largely overlapping settings. It would be helpful for readers to understand what is the additional insights, ideas, proofs, etc, in this present paper.
>
> The related work showed empirically that intermediate representations are equidistant on a straight line, i.e., RA1. However, they did not measure or even mention the remaining RA properties, i.e., RA2-4. In fact, they explicitly say in their work that they believe there is "layer-wise heterogeneity" in ResNets, i.e., they expect misalignment between residual blocks.
>
> Our research demonstrates empirically and theoretically, that such equidistant intermediate representations are the result of the Jacobian singular vectors aligning (RA2) and the top Jacobian singular values scaling inversely with depth (RA3-4) and, in fact, due to the absence of purported "layer-wise heterogeneity."
>
> We have incorporated the above ideas into our revised manuscript.
>
> >
>
> We are genuinely thankful to the reviewer for their queries. These led to changes that greatly improved the quality of our work. In light of these refinements, should you find it fitting, we would be most grateful for any potential reconsideration of our score.

---

### Official Review · Reviewer_7wWU · 2023-07-06

**Soundness:** 3 good
**Presentation:** 3 good
**Contribution:** 3 good
**Rating:** 6
**Confidence:** 2

**Summary:**

The paper tries to analyze the remarkable performance of ResNet architecture and they find the residual alignment phenomenon. The phenomenon is general and they also proposed a mathematical model call the Unconstrained Jacobian Models to theoretically analyze it.

**Strengths:**

The authors find an interesting phenomenon for residual networks called residual alignments. I think it can guide future exploration for module designing, model compression, and regularization techniques.

**Weaknesses:**

I think the weaknesses of this paper have they only discussed such a phenomenon. It will be great if they can utilize such phenomenon to design some useful techniques.

**Questions:**

1. Is such residual alignment phenomenon a general case in different models with skip connection, like ViT or ResGCN?
2. Are there any techniques or architectures can cause residual alignment and then lead to better performance besides skip connection?

---

> ### Author Rebuttal · Authors · 2023-08-10
>
> We express our sincere gratitude to the reviewer for their inspiring questions and careful review of our paper. Below, we respond to each point that was raised.
>
>
> > I think the weaknesses of this paper have they only discussed such a phenomenon. It will be great if they can utilize such phenomenon to design some useful techniques.
>
> We agree that building upon the identified phenomenon to devise practical techniques is a promising avenue for research. The primary focus of this paper was to shed light on the phenomenon and analyze it theoretically. While we see immense potential in its application, introducing and rigorously testing any new methodologies against established baselines would be beyond the scope of this study. Nonetheless, we look forward to future exploration and research on this front.
>
>
> > Is such residual alignment phenomenon a general case in different models with skip connection, like ViT or ResGCN?
>
> This is a wonderful question. We asked the exact same question ourselves as well. Addressing this question requires a precise definition of what Residual Jacobians are for such networks. For example, for a ViT, a Residual Jacobian could be defined for a single token or the whole sequence. Depending on the definition, one might or might not observe Residual Alignment. The same question applies to graph neural networks. We hope future work will shed light on this question since it would be extremely interesting to understand the pervasiveness of our observations.
>
>
> > Are there any techniques or architectures can cause residual alignment and then lead to better performance besides skip connection?
>
> Thank you for this thought-provoking question. Our experiments consistently revealed that architectures with skip connections were the only ones exhibiting residual alignment. Moving forward, we hope future research investigates whether Residual Alignment manifests in alternative networks. Such investigations are sure to contribute significantly to gaining a deeper understanding of this intriguing phenomenon and evaluating its prevalence. Moreover, we postulate existing regularization techniques, including layer permutation and structured dropout, should strengthen RA.
>
> To support this claim, the attached PDF showcases an additional experiment we conducted that illustrates how the incorporation of stochastic depth further boosts RA.
>
>
> >
>
> We would like to express our sincere appreciation to the reviewer for their insightful questions, which resulted in revisions that enhanced the quality of our work. In light of these refinements, should you find it fitting, we would be most grateful for any potential reconsideration of our score.

---

> ### Author Response · Authors · 2023-08-20
>
> Dear reviewer,
>
> As the rebuttal period is ending shortly, please let us know if you have any further questions or if we can provide further clarification.

---

### Official Review · Reviewer_EjRc · 2023-07-06

**Soundness:** 3 good
**Presentation:** 3 good
**Contribution:** 2 fair
**Rating:** 7
**Confidence:** 4

**Summary:**

This paper investigates the ResNet architecture, a popular deep-learning model known for its improved performance through skip connections. The authors aim to uncover the underlying mechanisms behind its success. They conduct an empirical study by linearizing the residual blocks of ResNet using Residual Jacobians and analyzing their singular value decompositions. The measurements and analysis conducted in the study reveal a phenomenon called Residual Alignment (RA), which is characterized by four fundamental properties. First, intermediate representations of a given input are evenly distributed on an embedded line in the space. Second, the top left and right singular vectors of Residual Jacobians align with each other and across different depths. Third, Residual Jacobians are at most rank C for fully-connected ResNets, where C represents the number of classes. Lastly, the top singular values of Residual Jacobians decrease inversely with depth. The study consistently demonstrates the occurrence of RA in ResNet models that generalize well. This phenomenon holds for both fully-connected and convolutional architectures across various depths and widths and different numbers of classes on benchmark datasets. However, RA ceases to occur when skip connections are removed. The authors also propose a novel mathematical model where RA is present. The findings suggest a strong alignment between residual branches in ResNet, imparting a rigid geometric structure to the intermediate representations as they progress linearly through the network until they reach the final layer, where Neural Collapse occurs.

**Strengths:**

Originality: Using the idea of linearization of the Residual Jacobian is commonly known for exploring the behavior of Residual Networks during the training and even pre-training. The authors use the idea of the Unconstrained Jacobian Model, which is interesting but also limited to a binary classification task. Still, it is novel to the best of my knowledge.

Quality: The motivation and idea are clearly defined, and experiments support the RA phenomena.

Clarity: The paper would benefit from improved organization, such as relocating the related work section from section 5 to section 3 or 2. Additionally, the appendix appears disorganized, with figures located randomly. Other than these, it is written very well and easy to follow.

Significance: The study's results demonstrate the presence of the Residual Alignment (RA) phenomenon in the singular value decomposition (SVD) of the linearized Residual Jacobian. The mathematical analysis further confirms the occurrence of RA in binary classification. However, the paper lacks a deeper exploration of RA (2-4) and other potential insights.


**Weaknesses:**

Linearization of the Residual Jacobian and analysis of its SVD decomposition: Despite being a valid idea, is not novel. The authors could also potentially investigate the distribution of singular values with the help of random matrix theory. These singular values depend on the input feature vectors (or hidden representations), but this dependency is not explored.

Limited applicability to binary classification: The use of Unconstrained Jacobian Models, while interesting, is deemed limited in the context of binary classification tasks. The paper does not explore its potential beyond this specific task.

Organization and clarity: The paper would benefit from improved organization, particularly in the placement of the related work section, which could be better situated in an earlier section. The appendix also lacks clear structure and organization, with figures placed randomly throughout.


**Questions:**

* It would have been great to demonstrate the relationship between RA and Neural Collapse a little bit more through experiments.

* Exploring the behavior of the singular values’ distribution can provide more insights into RA.


**Limitations:**

The authors do not provide potential limitations of their work.

---

> ### Author Rebuttal · Authors · 2023-08-10
>
> We express our sincere gratitude to the reviewer for the time and effort they put into the assessment of our paper as well as for their insightful comments and critiques. In what follows, we address each of the points raised.
>
> > Linearization of the Residual Jacobian and analysis of its SVD decomposition: Despite being a valid idea, is not novel.
>
> Thank you for bringing this to our attention. Indeed, [1] had previously proposed the linearization of residual Jacobians, and regrettably, we failed to recognize their contribution. We have now duly referenced their work in our manuscript.
>
> We would also like to emphasize that our novelty does not lie in the linearization of the residual Jacobians, as [1] had already explored this aspect. Instead, our original contribution centers around the discovery of the alignment between the residual layers, as well as its theoretical understanding.
>
> [1] Rothauge, K., Yao, Z., Hu, Z., & Mahoney, M. W. (2019). Residual networks as nonlinear systems: Stability analysis using linearization.
>
>
> > The authors could also potentially investigate the distribution of singular values with the help of random matrix theory. These singular values depend on the input feature vectors (or hidden representations), but this dependency is not explored.
>
> The authors of [1] have greatly advanced our understanding by thoroughly investigating the distribution of singular values in linearized residual Jacobians. Establishing a link between their findings and Residual Alignment represents an intriguing prospect that warrants a very careful and prolonged exploration. Moreover, Random Matrix Theory (RMT) stands out as a powerful tool capable of providing a formal framework to explore such connections. However, this endeavor extends considerably beyond the scope of our current work. We, therefore, leave it for future work.
>
>
> > Significance: The study's results demonstrate the presence of the Residual Alignment (RA) phenomenon in the singular value decomposition (SVD) of the linearized Residual Jacobian. The mathematical analysis further confirms the occurrence of RA in binary classification. However, the paper lacks a deeper exploration of RA (2-4) and other potential insights.
>
> We have enriched our manuscript with a more comprehensive exploration of RA by intertwining its attributes with established research findings. Specifically, we've connected RA to:
> 1. The stability analysis discussed in [1].
> 2. The concepts of Neural Collapse and layer-wise Neural Collapse.
> 3. The Law of Data Separation principle.
>
> Furthermore, the attached PDF includes two novel experiments that offer deeper insights into the RA phenomenon:
> 1. An experiment that showcases the simultaneous occurrence of Neural Collapse and Residual Alignment.
> 2. An investigation revealing that the introduction of stochastic depth amplifies the effects of RA.
>
>
> > Limited applicability to binary classification: The use of Unconstrained Jacobian Models, while interesting, is deemed limited in the context of binary classification tasks. The paper does not explore its potential beyond this specific task.
>
> Our goal was to develop a reasonable mathematical model that effectively demonstrates the emergence of Residual Alignment. We concur that expanding this theoretical framework to incorporate multi-class classification and other tasks presents an intriguing opportunity. Nonetheless, we defer a more rigorous, comprehensive exploration of these theoretical aspects to subsequent research.
>
>
> > Organization and clarity: The paper would benefit from improved organization, particularly in the placement of the related work section, which could be better situated in an earlier section. The appendix also lacks clear structure and organization, with figures placed randomly throughout.
>
> Thank you for your suggestion. We have now relocated the "Related Works" section from its previous position in Section 5 to its new place in Section 3. Furthermore, we have undertaken considerable efforts to refine the organization of the figures within the Appendix, ensuring they are arranged in a more meticulous and systematic manner.
>
>
> > It would have been great to demonstrate the relationship between RA and Neural Collapse a little bit more through experiments.
>
> We appreciate your wonderful recommendation. The enclosed PDF presents an experiment demonstrating the co-occurrence of Neural Collapse and Residual Alignment.
>
> >
>
> We again reiterate our appreciation to the reviewer for their detailed recommendations which led to revisions that improved the quality and clarity of our paper. In light of these refinements, should you find it fitting, we would be most grateful for any potential reconsideration of our score.

---

> > ### Comment · Reviewer_EjRc · 2023-08-17
> > **Response to authors' rebuttal**
> >
> > I thank the authors for their thorough responses. I think the revision significantly improves the quality of the paper.
> >
> > I think the paper is a nice contribution to the field, and have increased my score to Accept.

---

### Author Rebuttal · Authors · 2023-08-10

Enclosed, please find a page detailing further experiments we carried out in response to the reviewers' inquiries.

---

### Decision · Program_Chairs · 2023-09-21

**Decision:**

Accept (poster)

**Comment:**

The paper makes a significant and clear contribution toward understanding the role of residual connections in ResNets. As shown by well-designed experiments, they perform a specific form of iterative reasoning in which hidden representations are pushed along low-dimensional “conveyor belts”. Reviewers are in agreement that the paper should be accepted. It is my pleasure to also recommend acceptance.